# BUDDY: BUDGET-DRIVEN DYNAMIC DEPTH ROUTING FOR ADAPTIVE LARGE LANGUAGE MODELS INFERENCE

## ABSTRACT

Large language models require substantial computational resources for inference due to their massive number of parameters. Model layer pruning accelerates inference by eliminating redundant layers. However, existing layer pruning methods fail to meet users' flexible budget constraints and lack the ability to adaptively adjust the inference path. To address these issues, we propose **Buddy**, a budget-driven and adaptive inference framework. Specifically, we design a Decision Module that adaptively selects important layers to execute based on user input while satisfying a given budget constraint. Additionally, Buddy reuses the KV cache from the first layer and dynamically updates the context during inference, enabling adaptive adjustments to the inference path based on evolving contextual information. Furthermore, when no explicit budget is provided, a Budget Predictor automatically determines an appropriate inference cost to achieve an optimal trade-off between performance and computational efficiency. Extensive experiments on the Llama model demonstrate that Buddy consistently outperforms baseline methods under various pruning configurations.

## 1 INTRODUCTION

Large language models (LLMs) deliver strong performance across many tasks (e.g., text generation, language translation, and text summary (Chang et al., 2024)) but incur substantial latency and computational cost at inference time due to their growing parameters scale (Zhou et al., 2025). Depth-level pruning, which removes entire transformer layers, is recently a popular model compression method that reduces latency without modifying attention shapes.

Most existing depth-pruning methods are *static*: they estimate a global importance score for each layer and permanently remove a fixed subset prior to deployment (Kim et al., 2024; Song et al., 2024). Static pruning is inexpensive at runtime but fundamentally task-agnostic and cannot adapt to input difficulty or user constraints (Wee et al., 2025). Recent *dynamic* approaches improve flexibility by skipping layers on the fly (e.g., via routing (Jiang et al., 2024) or early-exit heuristics (Fan et al., 2024; Elhoushi et al., 2024)). However, they typically: (1) enforce a **fixed sparsity** pattern or weakly control the number of executed layers, which makes the actual computation unpredictable; and (2) determine a **fixed routing path** during the prefill stage and reuse it during decoding, even though the importance of layers can change as new tokens are generated.

In this paper, we target *adaptively pruning the model depth based on different inputs while also meeting the user's budget requirements*, enabling **a single deployed LLM to serve heterogeneous budgets.** However, achieving this requires two challenges: **(C1) Adaptive inference for inputs.** The importance of model layers varies greatly for different user inputs (Observation 2.2). Selecting model layers based on user input to execute the optimal inference path while meeting the user's budget poses the first challenge. **(C2) Adaptive inference during decoding.** In the autoregressive process, as the text is generated, the importance of the layer will change accordingly (Observation 2.2). However, often only one token is generated and taken as the next input, thus lacking global information for the path decision. The second challenge is how to dynamically select the path in both the prefill stage and the decode stage.

Table 1: A holistic comparison of Buddy with related approaches: static depth pruning and dynamic depth routing. Buddy provides comprehensive support for the all the properties.

| Method | Pruning | Budget Strict[2] | Budget Flexibility[3] | Input Adaptive[4] | Decode Adaptive[5] |
|---|---|---|---|---|---|
| Shortened Llama (Kim et al., 2024) | static | ✓ | ✗ | ✗ | ✗ |
| ShortGPT (Men et al., 2024) | static | ✓ | ✗ | ✗ | ✗ |
| SLEB (Song et al., 2024) | static | ✓ | ✗ | ✗ | ✗ |
| Early-exit (Elhoushi et al., 2024; Fan et al., 2024) | dynamic | ✗ | ✗ | ✓ | ✗ |
| Token-wise (Yang et al., 2025; Jiang et al., 2024) | dynamic | ✗ | ✗ | ✓ | ✓ |
| PuDDing (Wee et al., 2025) | dynamic | ✓ | ✗ | ✓ | ✗ |
| FiRST (Jain et al., 2025) | dynamic | ✗ | ✗ | ✓ | ✗ |
| **Buddy (Ours)** | dynamic | ✓ | ✓ | ✓ | ✓ |

We propose **Buddy**, a *budget-driven dynamic layer-selection* framework for LLMs. Given an input and an optional compute budget, Buddy selects a subset of layers to execute that (a) satisfies the budget constraint, and (b) preserves task performance. Concretely, to address the first challenge, Buddy introduces a lightweight *Decision Module* that scores intermediate layers conditioned on the current context and selects the top-$k$ layers to execute, where $k$ is tied to the user budget.[1] To stabilize learning and accelerate adaptation, we optionally initialize the Decision Module with *static priors* derived from standard importance indicators (e.g., perplexity-, Taylor-, or representation-similarity–based signals) after appropriate normalization and fusion.

To address the second challenge, Buddy introduces a mechanism to inject *global information during decoding*: we reuse the first layer's KV cache to summarize the global context and concatenate the newest token's features before each routing decision. This provides stable, low-overhead global signals to the Decision Module and enables Buddy to update the execution path as the generation unfolds.

Furthermore, when users omit a budget, Buddy turns to an *Adaptive Budget Predictor* that provides the optimal budget according to the inputs. In our experiments on the Llama family, Buddy achieves stronger accuracy–compute trade-offs than static and dynamic baselines at matched sparsity, while providing deterministic control over the executed compute.

Our contributions are summarized as:

- **Budget-aware decision making.** We design a lightweight Decision Module that scores layers conditioned on the current context and selects the top-$k$ layers to execute, providing *explicit* and *deterministic* control over compute under user-specified budgets.

- **Global information during decoding.** We introduce a low-overhead scheme that reuses the first layer's KV cache to expose global context to the router at every decoding step, enabling path updates as generation evolves.

- **Adaptive budget prediction.** When users do not specify a budget, a discrete Budget Predictor chooses a compute level that preserves task quality while minimizing executed layers.

- **Extensive empirical study.** On Llama models across multiple benchmarks, Buddy consistently outperforms static and dynamic baselines at the same sparsity, demonstrating effective budget control and robust quality.

## 2 BACKGROUND AND MOTIVATION

### 2.1 BACKGROUND

**LLMs Architecture.** Given a tokenized sequence $\mathcal{X} = \{x_1, x_2, \ldots, x_n\}$, an autoregressive LLM predicts the next token $x_{n+1}$ by applying an embedding layer, a stack of $L$ Transformer blocks, and

---

[1]We use "budget" to denote a discrete compute limit such as the number of executed layers per step.

[2]**Budget Strict:** strict to the user's budget constraint.

[3]**Budget Flexibility:** the same model meets users' diverse budgets.

[4]**Input Adaptive:** automatically adapts the inference path based on the input.

[5]**Decode Adaptive:** dynamically adjusting its decoding path during the generation process.

a classification head:

$$h_n \ = \ \mathcal{F}_L \circ \mathcal{F}_{L-1} \circ \cdots \circ \mathcal{F}_1 \circ \mathcal{F}_{\text{embed}}(\mathcal{X}), \qquad x_{n+1} \sim \text{Softmax}\big(\mathcal{F}_{\text{head}}(h_n)\big). \tag{1}$$

Here $\mathcal{F}_{\text{embed}}$ denotes token embeddings, $\{\mathcal{F}_\ell\}_{\ell=1}^L$ are Transformer blocks, and $\mathcal{F}_{\text{head}}$ maps the final hidden state to logits.

**Prefill vs. Decode.** Inference consists of a *prefill* phase and an *autoregressive decode* phase. In prefill, the model processes the entire prompt with teacher forcing. During decoding, the model generates one token at a time and appends it to the context. To avoid recomputing attention over the full history at every step, LLMs maintain a *KV cache* of keys/values from prior positions. Let $t$ be the current decoding step and $(Q_t, K_t, V_t)$ the query/key/value of the new token. With cached states $K_{\text{cache}}^{(t-1)}, V_{\text{cache}}^{(t-1)}$, attention uses and the cache is updated by:

$$\text{Attn}\big(Q_t, \ [K_{\text{cache}}^{(t-1)}; K_t], \ [V_{\text{cache}}^{(t-1)}; V_t]\big), \qquad K_{\text{cache}}^{(t)} = [K_{\text{cache}}^{(t-1)}; K_t], \qquad V_{\text{cache}}^{(t)} = [V_{\text{cache}}^{(t-1)}; V_t]. \tag{2}$$

This mechanism substantially reduces decoding compute.

**Layer Skipping.** Dynamic depth pruning realizes speedups by *skipping* selected Transformer blocks, enabled by residual connections. Let $\mathcal{M} \in \{0, 1\}^L$ be a binary skipping mask, where $\mathcal{M}_\ell = 1$ means "execute" and $\mathcal{M}_\ell = 0$ means "skip." With hidden states $\{\mathcal{H}_\ell\}_{\ell=0}^L$ (and $\mathcal{H}_0$ the input to the stack), the residual update is

$$\mathcal{H}_\ell = \begin{cases} \mathcal{F}_\ell(\mathcal{H}_{\ell-1}) + \mathcal{H}_{\ell-1}, & \text{if } \mathcal{M}_\ell = 1, \\ \mathcal{H}_{\ell-1}, & \text{if } \mathcal{M}_\ell = 0. \end{cases} \tag{3}$$

## 2.2 MOTIVATION

> ***Observation 1:*** *The importance of Transformer layers is* input-dependent*; different inputs induce different importance distributions across layers.*

**Input-Adaptive.** We compute per-layer importance of Llama2-7B on WikiText-2 (Merity et al., 2017) using the Taylor score and, for each input, sort layers to obtain a *remove order* (layers ranked earlier are less important and thus more removable). As illustrated in Figure 1, the resulting rank distributions vary markedly across inputs. For example, the 2nd layer may rank near 3rd on some inputs but around 22nd on others, indicating strong input dependence. This variability implies that the importance estimated once on a small validation set (static pruning) cannot accommodate diverse inputs; instead, pruning decisions should be made dynamically to adapt to different inputs.

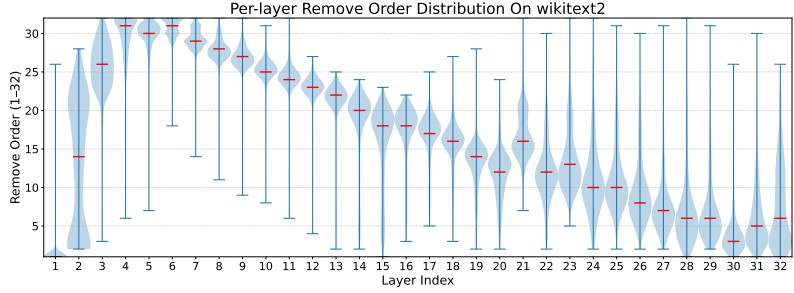

Figure 1: Input-dependent layer importance ranking distributions across different inputs on the WikiText-2 datasets. The chart shows how layer rankings vary significantly across inputs, demonstrating the necessity for dynamic pruning decisions.

> ***Observation 2:*** *The importance distribution evolves during decoding; as generation proceeds, the relative utility of layers changes with the growing context.*

**Decode-Adaptive.** To emulate different stages of autoregressive generation, we vary the context length from 256 to 512 and recompute layer importance. Figure 2 shows that as context length increases, the per-layer ranking of Llama2-7B shifts substantially. For example, the importance of the 14th layer of the model is relatively important at the beginning, then becomes unimportant, and then becomes important again during the decoding process. Hence, a routing path fixed at prefill is insufficient; pruning must adapt throughout decoding as new tokens extend the context.

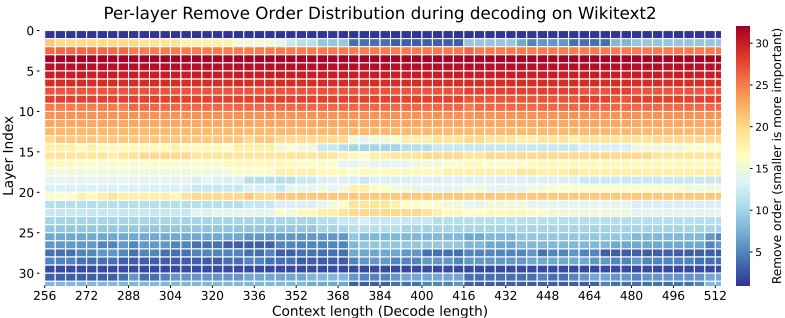

Figure 2: Evolution of layer importance rankings on the WikiText-2 dataset during autoregressive decoding as context length increases from 256 to 512 tokens. The heatmap illustrates how layer importance (e.g., layer 14) fluctuates throughout the decoding process, highlighting the need for decode-adaptive pruning strategies.

Based on the above two observations, we propose a dynamic layer execution method that can adapt to different inputs and select different layers for execution, while also dynamically modifying the inference path during the inference process.

## 3 METHOD

### 3.1 FRAMEWORK OVERVIEW

The framework overview of Buddy is summarized in Figure 3. Given a user prompt $x$ and an optional latency/compute budget $b$, Buddy selects a subset of Transformer blocks to execute. First, it encodes the input and extract *global context* from the KV states of the first layer (Section 3.3), and feeds it into a **Decision Module** (Section 3.2) that outputs a score $s_\ell$ for each block. Second, the Decision Module deterministically selects the top-$k$ blocks, which satisfy the budget $b$, and forms a binary layer mask $\mathcal{M} \in \{0,1\}^L$ for LLM execution. During autoregressive decoding, the global context is refreshed using the latest token and reused KV states, enabling the Decision Module to update the execution path as the generation evolves. Third, when the user does not specify a budget, a **Budget Predictor** (Section 3.4) infers a discrete compute level $\hat{b}$ from the same global context, aiming to minimize executed layers subject to maintaining task quality.

### 3.2 DECISION MODULE

#### 3.2.1 FORMULATION

Decision Module targets at choosing the optimal inference path for the given user input and budget, i.e., which layers are executed and which are skipped. For an LLM with $L$ layers $\mathcal{B} = \{\mathcal{B}_1, \ldots, \mathcal{B}_L\}$, there are $2^L$ possible paths, which makes exhaustive search infeasible. We therefore cast routing as a *ranking-and-selection* problem: rank layers by importance for the current input, then select the top-k layers to satisfy the budget $b$. Following common practice (Jiang et al., 2024), the first and the last blocks are always executed due to their outsized impact on quality. Let the number of middle layers be $L_{\text{set}} = L - 2$, and define the candidate set $\mathcal{B}_{\text{set}} = \{\mathcal{B}_2, \ldots, \mathcal{B}_{L-1}\}$.

The Decision Module contains a lightweight scorer $\mathcal{F}(\mathcal{H})$ (a lightweight MLP) that consumes the sequence representation $\mathcal{H} = \{\mathcal{H}_0, \ldots, \mathcal{H}_n\} \in \mathbb{R}^{n \times D}$ for $n$ tokens. We first extract features along the token dimension and normalize along the sequence, producing mid-layer scores $s = \{s_2, s_3, \ldots, s_{L-1}\} \in \mathbb{R}^{L_{\text{set}}}$. We then fuse $s$ with an offline prior $p \in \mathbb{R}^{L_{\text{set}}}$ (Section 3.2.2) and

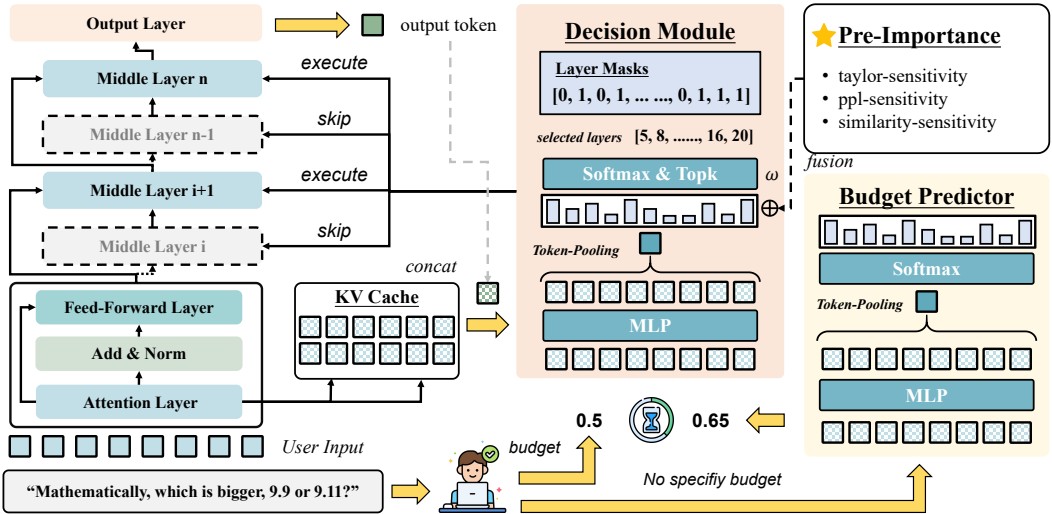

Figure 3: Overview of the Buddy framework. The framework consists of three key components: **(1) Decision Module** that adaptively selects layers based on context and budget constraints, **(2) KV-aware Planner** that reuses first-layer KV cache and updates global context during decoding, and **(3) Budget Predictor** that automatically provides optimal budgets when not explicitly provided.

apply a softmax to obtain normalized scores, from which we select the top layers subject to the budget. Let $k = \text{round}(b \cdot L) - 2$ be the number of middle layers to execute. The resulting path $\mathcal{P} = \{\mathcal{B}_i, \ldots, \mathcal{B}_k\} \in \mathbb{R}^k$ induces a binary mask $\mathcal{M} = \{m_i, \ldots, m_\ell\} \in \{0, 1\}^{L_{\text{set}}}$. The computation is formulated as:

$$s' = s + \omega \tilde{p}, \qquad \tilde{p} = \frac{p - \mu(p)}{\sigma(p) + \varepsilon}, \tag{4}$$

$$\mathcal{P}(s', b) = \text{Top-}K\big(\text{Softmax}(s'), k = \text{round}(b \cdot L) - 2\big), \tag{5}$$

where a small $\omega$ balances model-predicted scores and the prior. The middle layers then perform a single forward pass with dynamic depth[6]. Layer selection during inference can be viewed as a binary decision per layer (execute vs. skip): blocks with $m_\ell = 1$ are executed; blocks with $m_\ell = 0$ are skipped via identity routing while preserving residual alignment.

The Decision Module is trained jointly with the base model under the standard SFT objective. We pool the tokens, feed them into a two-layer MLP to obtain layer scores, convert the normalized budget into an integer k, and take the top-k scores to form a binary execution mask over intermediate layers. The mask gates each layer's residual update (executed if mask=1, skipped otherwise), and we adopt a straight-through estimator (STE) (Bengio et al., 2013) where the forward pass uses the hard mask while the backward pass uses a renormalized soft top-k mask so that gradients from the language modeling cross-entropy loss flow end-to-end into the gating mechanism. More details please refer to Appendix A.4.1

### 3.2.2 PRIOR NORMALIZATION AND FUSION

We compute offline layer priors on a small validation set (e.g., WikiText2 and PTB) , following prior work on depth pruning (Kim et al., 2024). We consider three types of indicators to initialize the prior (e.g., $\Delta$PPL when skipping layer $\ell$, Taylor/Fisher scores, and cosine-based dissimilarity). Because different indicators have different scales and units, we normalize them so that all priors are comparable and lie in $(0, 1)$. We first unify the direction so that "larger means more important" (for cosine, use $1 - \cos$). Next, we apply heavy-tail compression on nonnegative metrics via $\log(1 + x)$, then a robust $z$-score using the median and IQR:

$$z_\ell = \frac{\widetilde{x}_\ell - \text{median}_x}{\text{IQR}_x + \varepsilon}. \tag{6}$$

---

[6]We adopt the layer-skipping scheme described in 2.1)

Finally, we map to $(0, 1)$ using rank normalization (empirical CDF) across layers, yielding $\tilde{p}$. This unitless prior is then fused with the data-driven scores $s$ as in the formulation above. All priors are computed on the WikiText-2 validation set, which is used solely for offline importance estimation. Detailed procedures appear in Appendix A.3.

### 3.3 KV-AWARE PLANNER

During autoregressive inference, an LLM generates one token at a time. If the Decision Module were to consume only the newly generated token at each decoding step, it would lack sufficient context and thus struggle to choose an effective execution path. To address this, we *reuse the first layer's KV cache* to provide a lightweight global summary of the full prefix. Therefore, we concatenate the KV values of the newly generated tokens with those stored in the cache to obtain global contextual information, as described in Equation 2. We use $G_t \in \{K_t, V_t, K_t + V_t\}$ (discussed in the ablation study in Section 4.3) as the input to the Decision module, enabling dynamic prediction of the optimal reasoning path during the decoding stage.

### 3.4 BUDGET PREDICTOR

When the user does not provide a budget $b$, we predict an optimal $\hat{b}$ from the input. Since layer selection is performed at the *block* level, we discretize budgets into bins that represent *how many middle layers to execute*. Following the Decision Module formulation, we define the action space as the number of middle layers to execute: $\mathcal{A} = \{1, 2, \ldots, L_{\text{set}}\}$.

The **Budget Predictor** takes the same KV-aware context $G_t$ as input (Section 3.3) and outputs a categorical distribution $\pi_\theta(k \mid G_t)$ over the action space $\mathcal{A}$. The predicted layer count k and the corresponding budget are derived as:

$$\hat{k} = \arg\max_{k \in \mathcal{A}} \pi_\theta(k \mid G_t), \qquad \hat{b} = \frac{\hat{k} + 2}{L}. \tag{7}$$

where the "+2" accounts for the always-executed first and last layers. This budget is then fed to the Decision Module for layer selection, ensuring consistency with the ranking-and-selection framework.

We train the Budget Predictor using Group-Relative Policy Optimization (GRPO) (Shao et al., 2024) to optimize the trade-off between task performance and computational efficiency. The training objective balances predictive quality with computational cost through reinforcement learning. Details refer to Appendix A.4.2.

## 4 EXPERIMENTS

### 4.1 SETTINGS

**LLMs.** To demonstrate how Buddy performed on different models, we conduct experiments on Llama families (Touvron et al., 2023): Llama2 and Llama1. In particular, we fine-tune the 7B and 13B model with the specific versions detailed in Appendix A.6. We conduct experiments on removing 4, 8, 12, and 16 blocks of LLM, which indicates the pruning rate of 12.5%, 25%, 37.5%, and 50%

**Benchmark.** We conduct experiments for these LLMs on two different benchmarks. The first benchmark is **Commonsense Reasoning**, which includes BoolQ (Clark et al., 2019), PIQA (Bisk et al., 2020), HellaSwag (Zellers et al., 2019), WinoGrande (Sakaguchi et al., 2021), ARC-easy and ARC-challenge (Clark et al., 2018), OpenbookQA (Mihaylov et al., 2018), and SIQA (Sap et al., 2019). We employed lm-eval-harness (Gao et al., 2023) to create open prompts for the benchmarks and produce the results.

**Baselines.** We compare against recent depth-pruning approaches. *Static* methods: **(1) Shortened LLaMA** (Kim et al., 2024), rank blocks $\Delta$PPL ; **(2) ShortGPT** (Men et al., 2024), rank by input–output cosine similarity ; **(3) SLEB** (Song et al., 2024), iterative rank layers by $\Delta$PPL. *Dynamic* methods: **(4) PuDDing** (Wee et al., 2025), construct the omission set from the evaluated commonsense benchmarks; **(5) FiRST** (Jain et al., 2025), add lightweight per-layer linear routers to predict execute/skip decisions.

Table 2: Performance comparison on Llama2-7B across seven Common Sense Reasoning benchmarks for different pruning methods at various sparsity levels (12.5%-50%). **Boldface** indicates the best performance, and the Underline means the second-order performance. Buddy consistently outperforms baseline methods, achieving the highest average performance across all pruning ratios.

| Method | Pruning | RM Blocks | OBQA | PIQA | BoolQ | SIQA | Hellaswag | ARC-E | ARC-C | Winogrande | Avg. |
|---|---|---|---|---|---|---|---|---|---|---|---|
| Dense w/o | - | 0(0.0%) | 44.20 | 79.11 | 77.71 | 46.06 | 76.02 | 76.30 | 46.33 | 69.14 | 64.36 |
| Shortened Llama | static | 4(12.5%) | 40.60 | **78.02** | 71.25 | 45.29 | 72.53 | 73.27 | 42.83 | 62.90 | 60.84 |
| ShortGPT | static | 4(12.5%) | 42.00 | 77.15 | **78.69** | **47.59** | **73.69** | 72.56 | 44.45 | **69.14** | **63.16** |
| SLEB | static | 4(12.5%) | 41.40 | 77.48 | 71.96 | 45.14 | 72.60 | **73.40** | 41.04 | 64.01 | 60.88 |
| PuDDing | dynamic | 4(12.5%) | 40.00 | 75.95 | 73.55 | 43.30 | 69.73 | 69.32 | 39.51 | 64.48 | 59.48 |
| FiRST | dynamic | 4(12.5%) | 36.00 | 56.20 | 57.74 | 40.48 | 47.66 | 48.27 | 34.73 | 54.22 | 46.91 |
| Buddy | dynamic | 4(12.5%) | **43.00** | 77.09 | 73.30 | 47.34 | 73.64 | 73.02 | **44.80** | 68.82 | 62.63 |
| Shortened Llama | static | 8(25%) | 37.00 | **74.27** | 61.87 | 41.10 | 63.15 | 63.26 | 34.90 | 54.62 | 53.77 |
| ShortGPT | static | 8(25%) | 39.60 | 72.42 | 62.94 | 43.71 | **67.66** | 65.28 | **38.91** | **67.09** | 57.20 |
| SLEB | static | 8(25%) | **39.80** | 73.78 | 69.24 | 43.19 | 65.72 | **66.58** | 35.49 | 60.46 | 56.78 |
| PuDDing | dynamic | 8(25%) | 36.60 | 71.82 | 62.87 | 39.30 | 60.30 | 61.41 | 35.58 | 56.67 | 53.07 |
| FiRST | dynamic | 8(25%) | 35.40 | 55.22 | 57.74 | 40.38 | 44.52 | 45.20 | 32.94 | 40.38 | 43.97 |
| Buddy | dynamic | 8(25%) | 39.20 | 73.18 | **72.63** | **45.65** | 66.64 | 65.28 | 38.05 | 66.85 | **58.44** |
| Shortened Llama | static | 12(37.5%) | 34.20 | 70.89 | 62.14 | 39.87 | 52.81 | 57.28 | 29.86 | 52.33 | 49.92 |
| ShortGPT | static | 12(37.5%) | 33.20 | 66.92 | 71.22 | **43.65** | **58.76** | 54.46 | **34.13** | **64.09** | 53.31 |
| SLEB | static | 12(37.5%) | **36.00** | **71.38** | 60.28 | 41.15 | 54.83 | **59.76** | 31.06 | 52.96 | 50.93 |
| PuDDing | dynamic | 12(37.5%) | 31.60 | 64.69 | 46.42 | 37.36 | 47.36 | 49.16 | 29.35 | 53.59 | 44.94 |
| FiRST | dynamic | 12(37.5%) | 36.20 | 55.17 | 55.87 | 39.82 | 43.99 | 45.33 | 30.89 | 53.67 | 45.12 |
| Buddy | dynamic | 12(37.5%) | 35.60 | 70.78 | **72.81** | 41.81 | 57.87 | 58.08 | 31.14 | 60.22 | **53.54** |
| Shortened Llama | static | 16(50%) | 30.80 | **65.51** | 62.17 | **39.76** | 35.46 | **49.71** | 26.62 | 51.30 | 45.17 |
| ShortGPT | static | 16(50%) | 29.80 | 61.81 | **62.20** | 39.00 | 47.13 | 44.91 | 29.18 | **57.22** | 46.41 |
| SLEB | static | 16(50%) | 31.00 | 65.07 | 61.71 | 38.84 | 42.93 | 49.49 | 26.37 | 52.57 | 46.00 |
| PuDDing | dynamic | 16(50%) | 31.60 | 64.47 | 46.42 | 37.36 | 47.36 | 49.16 | **29.35** | 53.59 | 44.91 |
| FiRST | dynamic | 16(50%) | **33.20** | 53.21 | 54.53 | 32.91 | 39.73 | 41.12 | 29.27 | 53.28 | 42.15 |
| Buddy | dynamic | 16(50%) | 32.80 | 64.74 | 60.21 | 37.92 | **48.66** | 48.91 | 28.41 | 53.51 | **46.90** |

Table 3: Speed analysis on the Alpaca and SamSum datasets. The speed is measured as tokens/s.

| Method | rm blocks (ratio) | Alpaca | | | | SamSum. | | | |
|---|---|---|---|---|---|---|---|---|---|
| | | Prefill | Speed up | Decode | Speed Up | Prefill | Speed Up | Decode | Speed Up |
| Dense | 0(0.0%) | 1753.16 | ×1.00 | 39.48 | ×1.00 | 4033.83 | ×1.00 | 60.70 | ×1.00 |
| Buddy | 4(12.5%) | 1991.54 | ×1.14 | 40.03 | ×1.01 | 4035.68 | ×1.00 | 61.28 | ×1.01 |
| Buddy | 8(25%) | 2440.14 | ×1.39 | 46.80 | ×1.19 | 4777.07 | ×1.18 | 71.91 | ×1.18 |
| Buddy | 12(37.5%) | 2548.39 | ×1.45 | 51.49 | ×1.30 | 5476.04 | ×1.36 | 79.11 | ×1.30 |
| Buddy | 16(50%) | 3348.18 | ×1.91 | 64.84 | ×1.64 | 6581.55 | ×1.63 | 99.12 | ×1.63 |

**Hyper-parameters and Training Details.** We adopted the popular LoRA ($r = 8$, all linear modules) method to fine-tune the LLMs. We used Alpaca (Taori et al., 2023) [7], the comprehensive instruction tuning dataset, to fine-tune the models. We applied the $V - State$ as the features and adopted the taylor metric as the prior-knowledge, and set the $\lambda$ factor to 0.1. We converted the model precision to BFloat16 and used AdamW as the optimizer with 100 warm-up steps and trained the model with a learning rate of $1 \times 10^{-4}$ and batch size 8 for 2 epochs. We apply the same training configuration across each baseline. During training, the budget is generated randomly from each of the sparsity candidates.

**Main Results.** Across seven common-sense reasoning benchmarks and four sparsity settings, **Buddy** exhibits strong average performance and robust degradation as pruning deepens from Table 2. At $12.5\%$ sparsity (removing 4 blocks), Buddy attains an average of **62.63**, ranking second and trailing the best baseline (ShortGPT, 63.16) by only 0.53 points. For higher sparsities, Buddy delivers the best average in every case: **58.44** at $25\%$ ($+1.24$ over the best baseline), **53.54** at $37.5\%$ ($+0.23$), and **46.90** at $50\%$ ($+0.49$). Relative to the dense (unpruned) model average of 64.36, Buddy preserves $97.3\%$, $90.8\%$, $83.2\%$, and $72.9\%$ of accuracy at 12.5%, 25%, 37.5%, and 50% sparsity, respectively.

Buddy delivers the best *average* accuracy at moderate-to-high sparsity ($\geq 25\%$) while remaining highly competitive at light sparsity. Moreover, its budget flexibility makes it adapt to different sparsity ratios using only one model.

---

[7] https://huggingface.co/datasets/yahma/alpaca-cleaned

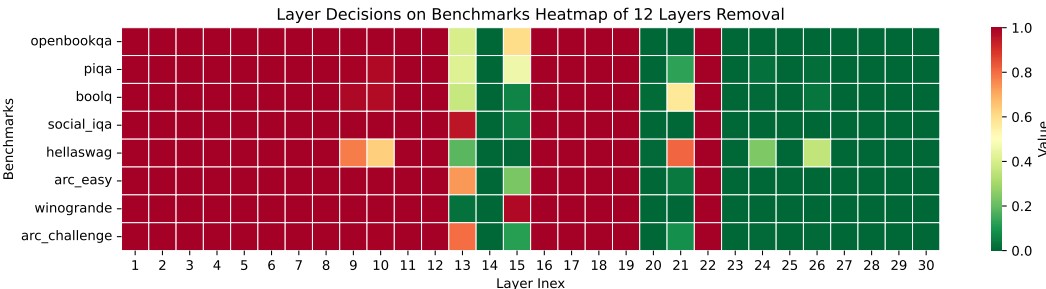

Figure 4: A visual illustration of the Buddy's average decision of each transformer block on 37.5% sparsity (12 layers removed). The color red indicates that the blocks are likely to be important, and the color green indicates that the blocks are likely to be pruned.

## 4.2 ANALYSIS

### 4.2.1 SPEED ANALYSIS

We measure end-to-end throughput (tokens/s) on ALPACA and SAMSUM for both *prefill* and *decode* phases. During decoding, the model generates 128 tokens. Results show in Table 3, indicating that (1) Speedups are consistently larger in prefill than in decode; (2) ALPACA exhibits larger prefill gains than SAMSUM at matched sparsity. (3) Gains increases with sparsity. At very light pruning (12.5%), the routing overhead offsets most of the savings, especially on SAMSUM; beyond $\sim 25\%$ sparsity, benefits clearly outweigh overhead.

### 4.2.2 LAYER SELECTION ANALYSIS

We analyze routing decisions across tasks and inputs. Figure 4 summarizes, at a 37.5% sparsity setting, the per-layer *execution frequency* averaged over inputs from eight benchmarks. The pattern reveals several *consistently important* layers—$[1:8]$ and $[16:17]$—that are almost always executed, and several *consistently unimportant* layers—$\{14, 23, 25, 27\text{–}30\}$—that are frequently pruned across tasks. The remaining layers adapt to the task: for example, layers $\{13, 15, 21\}$ show large variability, being crucial for some datasets but dispensable for others. These findings support input- and task-adaptive routing. Additional analyses under other sparsity levels appear in the Appendix.

### 4.2.3 DECODE ADAPTIVE ANALYSIS

**Inference path counts.** We examine how often the execution path changes during decoding. For ALPACA and SAMSUM, we run inference at multiple sparsity levels and count the number of distinct layer masks observed across decoding steps. As shown in Figure 5a, path changes are rare at low sparsity (remove 4 blocks), indicating that light pruning yields a largely stable route. At moderate sparsity, the number of distinct paths increases substantially, reflecting stronger context-driven adaptation. The effect is more pronounced on ALPACA than on SAMSUM, consistent with longer prompts and richer context evolution in ALPACA.

**Performance comparison.** We explore whether path recomputation during decoding helps accuracy. We compare two strategies on SAMSUM using ROUGE (Lin, 2004): *Reuse*, which fixes the decode path to the prefill decision, versus *Recompute*, which updates the path at each step using fresh hidden states. As in Figure 5b, *Recompute* consistently outperforms *Reuse* on ROUGE-1 and ROUGE-L, indicating that adapting the route to the evolving context yields better summaries without increasing the executed-layer budget.

### 4.3 ABLATION STUDY

**Hidden-State Features.** We compare three inputs to the Decision Module extracted from the first-layer KV cache: (1) Key states, (2) Value states, and (3) Key+Value element-wise plus. As shown in Table 4, using *Value* states yields the best accuracy at light/moderate pruning (12.5%–37.5%), while at extreme sparsity (50%) *Key* states slightly edge out Values. The *Key+Value* concatenation underperforms, suggesting redundancy and harder calibration for the scorer.

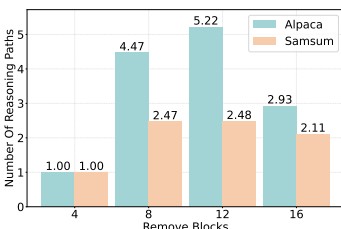 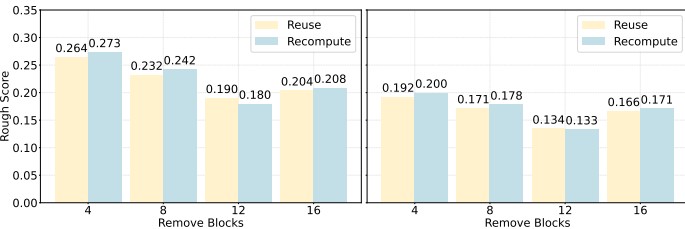

(a) Inference path count.    (b) ROUGE-1 (left) and ROUGE-L (right) between Reuse/Recompute.

**Prior Knowledge Fusion.** We ablate the layer-importance priors used in score fusion: (1) None, (2) $\Delta$PPL, (3) Cosine dissimilarity, and (4) Taylor/Fisher. Results in Table 5 show **Taylor** consistently performs best across pruning ratios. $\Delta$PPL lags at mild pruning but becomes competitive at high sparsity (50%). Overall, strong priors help, especially when compute is tight.

**Omega Coefficient.** We conducted ablation experiments on the $\omega$ hyperparameter in the Decision Module under different pruning ratios, testing six different lambda values: 0.01, 0.1, 0.3, 0.5, 0.7, and 1.0. As shown in Table 6, $\omega$=0.1 achieved the best performance across all pruning settings. In addition, we found that as $\omega$ increases, performance gradually decreases, meaning that prior knowledge fusion only requires a small amount of guidance, and excessive participation will lead to performance degradation.

Table 4: Ablation study on Decision Module input features using different hidden state representations. Average accuracy (%) is reported.

Table 5: Ablation study on prior knowledge integration for score fusion in the Decision Module. Average accuracy (%) is reported.

| rm blocks (ratio) | Key States | Value States | Key+Value States |
|---|---|---|---|
| 4(12.5%) | 62.12 | **62.21** | 60.92 |
| 8(25.0%) | 55.71 | **58.32** | 55.27 |
| 12(37.5%) | 50.10 | **52.44** | 49.34 |
| 16(50.0%) | **45.15** | 44.78 | 43.67 |

| rm blocks (ratio) | None | $\Delta$PPL | Cosine Similarity | Taylor |
|---|---|---|---|---|
| 4(12.5%) | 62.21 | 60.54 | 62.20 | **62.63** |
| 8(25.0%) | 58.32 | 55.03 | 58.27 | **58.44** |
| 12(37.5%) | 52.44 | 49.98 | 53.27 | **53.54** |
| 16(50.0%) | 44.78 | 46.45 | 45.93 | **46.90** |

Table 6: Ablation on the distillation coefficient $\omega$. Average accuracy (%) is reported.

| rm blocks (ratio) | $\omega$=0.01 | $\omega$=0.1 | $\omega$=0.3 | $\omega$=0.5 | $\omega$=0.7 | $\omega$=1.0 |
|---|---|---|---|---|---|---|
| 4 (12.5%) | 62.24 | **62.63** | 62.09 | 62.07 | 61.84 | 61.57 |
| 8 (25.0%) | 57.94 | **58.44** | 57.90 | 57.89 | 57.68 | 56.99 |
| 12 (37.5%) | 48.83 | **53.54** | 52.17 | 51.38 | 44.48 | 53.18 |
| 16 (50.0%) | 42.52 | **46.90** | 43.03 | 34.85 | 35.30 | 34.61 |

## 5    RELATED WORK

We review depth sparsity for Transformer LLMs and position our framework relative to static pruning and dynamic layer selection. A concise summary appears in Table 1.

### 5.1    STATIC DEPTH PRUNING

Depth pruning removes redundant Transformer blocks to reduce inference latency and memory (Wang et al., 2024). Recent studies typically score layer importance and drop low-scoring blocks. Methods such as ShortGPT (Men et al., 2024) and Shortened LLaMA (Kim et al., 2024) rank blocks by Block Influence (cosine), Taylor score, or $\Delta$PPL. Iterative schemes such as SLEB (Song et al., 2024) further identify redundancy and eliminate low-impact layers progressively. While effective, static pruning is task-agnostic and is fixed post-calibration, and cannot adapt to input difficulty or enforce user-specified compute budgets.

### 5.2    DYNAMIC DEPTH PRUNING

Dynamic methods decide at inference which layers to execute based on the current input or context. *early exit* methods (Fan et al., 2024; Elhoushi et al., 2024) produce predictions from intermediate

layers and skip the rest, but may degrade quality on difficult instances by discarding later computations. *layer-skipping* methods aim to preserve the depth of reasoning while avoiding redundant blocks via designing routers to skip computation based on token-level (Jiang et al., 2024; Raposo et al., 2024; Yang et al., 2025; Luo et al., 2025) or prompt-level (Wee et al., 2025; Jain et al., 2025) signals. Furthermore, PuDDing (Wee et al., 2025) trains a prompt-conditioned global router that selects an omission set of blocks before decoding. Although dynamic pruning adapts to inputs, most prior work can not strictly conform to the budget and must train separate models for different sparsity targets. Our work exposes budget as a first-class constraint and learning a router that honors user-specified (or predicted) budgets while updating paths throughout autoregressive decoding.

## 6 CONCLUSION

In this paper, we introduce Buddy, a budget-driven, decode-adaptive depth routing framework for LLM inference. A Decision Module automatically selects the most suitable layers for inference based on user input and budget constraints. A KV-aware planner reuses the KV cache to update the model's inference path during the autoregressive inference stage. A Budget Predictor automatically determines the optimal budget based on user input when no explicit budget is provided. The framework updates routes during decoding and honors explicit or predicted budgets. Experiments on Llama show consistent gains over static and dynamic baselines at matched sparsity.

## 7 ETHICS STATEMENT

This study builds upon publicly available pre-trained large language models, including Llama2-7B and Llama1-13B, and utilizes existing instruction-tuning datasets such as Alpaca. No new datasets were collected, and no human annotators were involved in the research process. As such, this work does not introduce any additional ethical concerns beyond those already associated with the use of large-scale pre-trained models and standard natural language processing benchmarks. All experiments were conducted in accordance with the ethical guidelines and terms of use of the employed models and datasets.

## 8 REPRODUCIBILITY STATEMENT

We have thoroughly elucidated our design and training details throughout the paper to provide a comprehensive understanding of our methodology. Specifically, the overall model architecture is described in Section 3, where we outline the overall design and describe the sub-modules in detail. We implement our method in PyTorch using HuggingFace Transformers and PEFT, and train/fine-tune base LLM checkpoints with LoRA. All datasets, preprocessing steps, training hyperparameters (including LoRA rank, optimizer, learning rate schedule, batch size, and epochs) 4.1. Evaluation follows the open-prompt settings in `lm-eval-harness` on the listed benchmarks, and throughput is measured with a fixed generation length under the hardware configurations described. Furthermore, the hardware and environment used for our experiments are specified in Appendix A.6, including the specifications of the computing resources and software versions that facilitated our experiments. We also list the specific versions of the large language models (LLMs) utilized in Appendix A.7, ensuring clarity regarding the models' configurations. We include scripts/configs to reproduce the accuracy across sparsity budgets and the reported speedups. providing insight into the settings that governed the training process. By compiling these details, we aim to enhance the reproducibility of our work and assist other researchers in understanding and applying our methods effectively.

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

## A    APPENDIX

### A.1    USE OF LLMS

In this study, LLMs are employed as automated proofreading agents to inspect, revise, and polish the manuscript. Specifically, they are tasked with detecting spelling and grammatical errors, assessing logical coherence and fluency of expression, and rewriting or condensing paragraphs where necessary.

### A.2    OBSERVATIONS

We also adopt $\delta$PPL as the evaluation metric on the LLaMA2-7B model, and compute the layer-importance distributions on the WikiText-2 and PTB datasets. The results are shown in Figure 6 and 7. They reveal that the estimated importance of different layers varies dramatically when different metrics are used (Taylor score vs. $\Delta$PPL). Moreover, the importance of each layer also differs across evaluation datasets (WikiText-2 vs. PTB). These phenomena further corroborate Observation 2.2: as the input and evaluation metric change, the importance of model layers also shifts, which calls for a dynamic layer-pruning scheme that can adapt to different inputs.

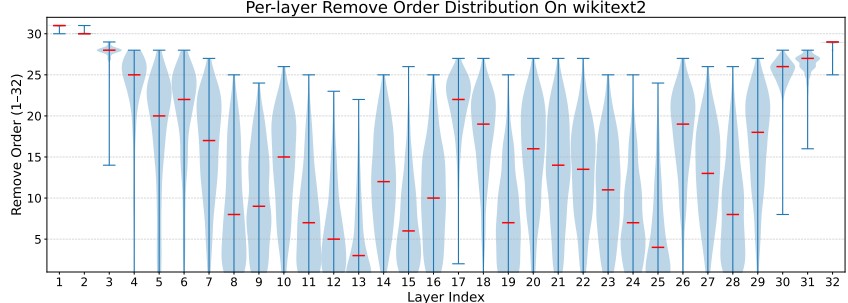

Figure 6: Input-dependent layer importance ranking distributions across different inputs on the WikiText-2 datasets. The chart shows how layer rankings vary significantly across inputs, demonstrating the necessity for dynamic pruning decisions.

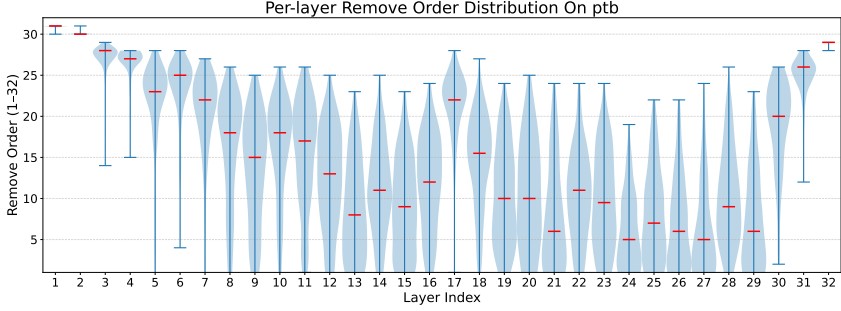

Figure 7: Input-dependent layer importance ranking distributions across different inputs on the PTB datasets. The chart shows how layer rankings vary significantly across inputs, demonstrating the necessity for dynamic pruning decisions.

### A.3    PRIOR INDICATORS AND NORMALIZATION

We consider $L$ transformer layers and $M$ prior indicators that estimate per-layer importance. Let $x_{l,m}$ denote the raw score of indicator $m \in \{1, \ldots, M\}$ on layer $l \in \{1, \ldots, L\}$. Different indicators can have heterogeneous scales and tails (e.g., perplexity gaps versus gradient norms), so we map them to a common, robust scale before fusion. Our pipeline has four steps; each step is applied per indicator $m$ across layers $l$.

**(i) Direction unification.** Ensure "larger $\Rightarrow$ more important" for every indicator:

- **$\Delta$PPL**: $x_{l,\text{ppl}} := \max\{0,\ \text{PPL}_{\text{skip}(l)} - \text{PPL}_{\text{full}}\}$.
- **Taylor/Fisher/GradNorm**: already positively oriented; clip to nonnegative.
- **Cosine similarity**: convert to *dissimilarity* $d_{l,\cos} := 1 - \cos(\cdot, \cdot) \in [0, 2]$ and set $x_{l,\cos} := d_{l,\cos}$.

**(ii) Heavy-tail stabilization.** For heavy-tailed indicators (typically $\Delta$PPL and Taylor/Fisher/Grad-Norm), apply a monotone variance–stabilizing transform, e.g.,

$$\tilde{x}_{l,m} \;=\; \log(1 + x_{l,m}) \quad (\text{or Box–Cox: } \tilde{x}_{l,m} = \tfrac{x_{l,m}^{\lambda} - 1}{\lambda},\ \lambda \in [0.2, 0.5]). \tag{8}$$

For cosine dissimilarity, we typically use the identity: $\tilde{x}_{l,\cos} := x_{l,\cos}$.

**(iii) Robust standardization and outlier control.** Compute a robust $z$-score using median and interquartile range (IQR) across layers:

$$z_{l,m} \;=\; \frac{\tilde{x}_{l,m} - \text{median}_l(\tilde{x}_{l,m})}{\text{IQR}_l(\tilde{x}_{l,m}) + \varepsilon}, \qquad \text{IQR} := Q_{75} - Q_{25},\ \varepsilon > 0. \tag{9}$$

Optionally winsorize: $z_{l,m} \leftarrow \text{clip}(z_{l,m}, -c, c)$ with $c \in [2.5, 3.5]$.

**(iv) Unit-interval mapping.** Map $z_{l,m}$ to a comparable, dimensionless score $u_{l,m} \in (0, 1)$. We use rank normalization (scale-free and robust):

$$u_{l,m} \;=\; \frac{1 + \text{rank}_l(z_{l,m})}{L + 1} \quad \in (0, 1), \tag{10}$$

or, alternatively, a temperature-controlled sigmoid $u_{l,m} = \sigma(z_{l,m}/\tau)$ with $\tau \in [1, 2]$.

The above yields $M$ normalized importance profiles $\{u_{\cdot,m}\}_{m=1}^{M}$.

### A.3.1 INDICATOR-SPECIFIC RECIPES

**A. $\Delta$PPL (skip-induced perplexity increase).**

$$\Delta\text{PPL}_l = \max\{0,\ \text{PPL}_{\text{skip}(l)} - \text{PPL}_{\text{full}}\}, \tag{11}$$

$$u_{l,\text{ppl}} = \text{RankNorm}\Big(\text{RobustZ}\big(\log(1 + \Delta\text{PPL}_l)\big)\Big). \tag{12}$$

This emphasizes layers whose removal most degrades language modeling quality while suppressing magnitude outliers.

**B. Taylor/Fisher/gradient-norm importance.**

$$u_{l,\text{taylor}} = \text{RankNorm}\Big(\text{RobustZ}\big(\log(1 + \text{Score}_l^{\text{taylor}})\big)\Big). \tag{13}$$

Any nonnegative saliency-like score (e.g., first-order Taylor, Fisher diagonal, $\ell_2$ gradient norm) fits this template.

**C. Cosine-similarity based discrepancy.** Let $s_l^{\text{full}}$ and $s_l^{\text{skip}(l)}$ be per-layer summary representations (or outputs) from the full and "skip-$l$" forward passes, respectively. Define the cosine dissimilarity $d_l = 1 - \cos\big(s_l^{\text{skip}(l)},\ s_l^{\text{full}}\big)$ and set

$$u_{l,\cos} = \text{RankNorm}\Big(\text{RobustZ}(d_l)\Big). \tag{14}$$

This treats larger representational deviation as higher importance.

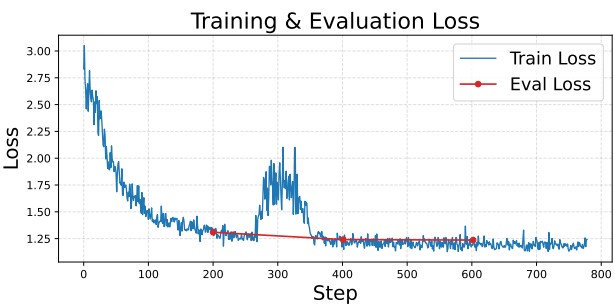

Figure 8: Training and validation loss curve of Buddy.

## A.4 IMPLEMENTATION

### A.4.1 DECISION MODULE

**Training Details and Overhead**

We train the Decision Module using standard supervised fine-tuning (SFT) jointly with the backbone model and LoRA adapters. Concretely, we fine-tune on the Alpaca instruction-following dataset using the Adam optimizer, and minimize the next-token cross-entropy loss. During this process, the parameters of the Decision Module (a lightweight MLP scorer) and the LoRA parameters of the backbone are updated simultaneously, following the same optimization hyperparameters as in the main LoRA fine-tuning setup described in Section 4.1. To make the discrete layer-selection decisions trainable, we adopt a straight-through estimator (STE). In the forward pass, the Decision Module produces per-layer scores, and we apply a hard Top-$k$ gating to obtain binary execute/skip decisions. In the backward pass, we use the continuous scores to approximate the gradient, allowing gradients from the language modeling loss to flow through the gating mechanism and update the MLP parameters.

In order to enable the model to both strictly and flexibly adapt to different compute budgets, we perform *random budget sampling* during training. For each mini-batch, we sample target sparsity levels from the same range used at inference time, and assign (potentially different) budgets to individual examples within the batch. This encourages the Decision Module to learn routing policies that are robust across a variety of budgets instead of overfitting to a single sparsity setting. We also explicitly include the zero-sparsity case (i.e., all layers executed) in the sampling space, so that every Transformer block is regularly activated during training and receives gradient updates, which improves overall training stability and robustness.

Figure 8 shows the training and validation loss curves of Buddy. The loss decreases smoothly throughout training, indicating that jointly optimizing the backbone, LoRA adapters, and the Decision Module is stable under our STE-based routing. Under the configuration in Section 4.1, training Buddy with the Decision Module consumes approximately 51.38 GB of GPU memory, while standard LoRA fine-tuning under the same setting uses about 49.72 GB. Thus, the additional memory overhead of the Decision Module during training is only around 1.66 GB compared to a vanilla LoRA run.

**Inference overhead**

We first analyze the theoretical latency of the Decision Module. We denote by $d$ the hidden size, by $d_c$ the MLP intermediate size (for LLaMA-7B, $d_c = 11008$ when $d = 4096$), by $T$ the input length, by $L$ the total number of transformer layers, by $I$ the number of ignored (fixed) layers in the router, by $r$ the hidden width of the Decision Module MLP, by $B$ the batch size, and by $b \in (0, 1]$ the budget parameter that controls the effective fraction of active layers. The Decision Module takes an input $x \in \mathbb{R}^{B \times T \times d}$ and applies two linear layers token-wise, $d \to r$ and $r \to (L - I)$, followed by a softmax and Top-$k$ whose FLOPs are negligible compared to the dense matrix multiplications. Counting one multiply–add as 2 FLOPs, its cost is approximated by

$$F_{\text{DM}} \approx 2BTdr + 2BTr(L - I).$$

For the backbone transformer, let $F_{\text{layer}}$ denote the FLOPs of a single decoder layer with hidden size $d$ and sequence length $T$. A LLaMA-style decoder layer has: three projections $Q, K, V$ with cost $3 \times 2Td^2 = 6Td^2$, one output projection with cost $2Td^2$, attention score/product $QK^\top$ and $AV$ with cost approximately $4T^2d$, and an MLP with architecture $d \to d_c \to d$, whose two linear layers cost $2Tdd_c + 2Td_cd = 4Tdd_c$. Thus the corrected per-layer FLOPs with $d \to d_c \to d$ are

$$F_{\text{layer}} \approx (6Td^2 + 2Td^2) + 4T^2d + 4Tdd_c = 8Td^2 + 4T^2d + 4Tdd_c.$$

Executing all $L$ layers (no skipping) costs

$$F_{\text{full}} \approx LF_{\text{layer}} = L\big(8Td^2 + 4T^2d + 4Tdd_c\big).$$

The router converts the budget $b$ into a Top-$k$ via $k(b) = bL - I$ (clamped to $[1, L - I]$), so that roughly $bL$ out of the $L$ layers are effectively active (the $I$ ignored layers are always run and about $k(b)$ of the remaining $L - I$ layers are selected). Accordingly, the backbone FLOPs under budget $b$ are approximated as

$$F_{\text{model}}(b) \approx bF_{\text{full}} = bL\big(8Td^2 + 4T^2d + 4Tdd_c\big).$$

The relative FLOPs overhead of the Decision Module at budget $b$ is then

$$\rho(b) \approx \frac{F_{\text{DM}}}{F_{\text{DM}} + F_{\text{model}}(b)} \approx \frac{F_{\text{DM}}}{bL\big(8Td^2 + 4T^2d + 4Tdd_c\big)},$$

where the second approximation uses that $F_{\text{DM}} \ll F_{\text{model}}(b)$ in practice.

Instantiating with the configuration $L = 32$, $d = 4096$, $d_c = 11008$ (Llama-7B MLP), $T = 1024$, $r = 32$, $I = 2$, $B = 1$, we obtain for the Decision Module $F_{\text{DM}} \approx 2 \cdot 1 \cdot 1024 \cdot 4096 \cdot 32 + 2 \cdot 1 \cdot 1024 \cdot 32 \cdot (32 - 2) = 270{,}401{,}536 \approx 2.70 \times 10^8$. For a single transformer layer the corrected cost is $F_{\text{layer}} \approx 8 \cdot 1024 \cdot 4096^2 + 4 \cdot 1024^2 \cdot 4096 + 4 \cdot 1024 \cdot 4096 \cdot 11008 \approx 339{,}302{,}416{,}384 \approx 3.39 \times 10^{11}$. Under budget $b = 0.875$, the backbone FLOPs are

$$F_{\text{model}}(0.875) \approx 0.875 \cdot F_{\text{full}} \approx 9{,}500{,}467{,}658{,}752 \approx 9.50 \times 10^{12},$$

giving a Decision Module FLOPs ratio of

$$\rho(0.875) \approx \frac{F_{\text{DM}}}{F_{\text{model}}(0.875)} \approx \frac{2.70 \times 10^8}{9.50 \times 10^{12}} \approx 2.85 \times 10^{-5} \approx 0.00285\%.$$

Under budget $b = 0.5$, the backbone FLOPs become

$$F_{\text{model}}(0.5) \approx 0.5 \cdot F_{\text{full}} \approx 5{,}428{,}838{,}662{,}144 \approx 5.43 \times 10^{12},$$

leading to

$$\rho(0.5) \approx \frac{F_{\text{DM}}}{F_{\text{model}}(0.5)} \approx \frac{2.70 \times 10^8}{5.43 \times 10^{12}} \approx 4.98 \times 10^{-5} \approx 0.00498\%.$$

Therefore, the Decision Module's routing overhead is negligible compared to the main transformer."'

### A.4.2 TRAINING WITH GRPO

**Problem Setup.** Given an input $x$, we extract the KV-aware context $G = \phi(x)$ using the same feature extraction as described in Section 3.3. The Budget Predictor is a categorical policy $\pi_\theta(k \mid G)$ over the action space $\mathcal{A} = \{1, 2, \ldots, L_{\text{set}}\}$, where action $k$ corresponds to executing $k$ middle layers. All other model parameters remain frozen during training.

**Reward Function.** For a target sequence $y$, we run the frozen model under budget $b(k) = \frac{k+2}{L}$ in teacher-forcing mode to obtain token probabilities $\hat{p}_{b(k)}$. The per-sample cross-entropy loss is:

$$\text{CE}\big(\hat{p}_{b(k)}, y\big) = -\frac{1}{T - 1} \sum_{t=2}^{T} \log \hat{p}_{b(k)}(y_t \mid x, y_{<t}). \tag{15}$$

We balance predictive performance and computational cost via:

$$r(k) = -\lambda_{\text{perf}} \cdot \text{CE}\big(\hat{p}_{b(k)}, y\big) - \lambda_{\text{cost}} \cdot b(k), \tag{16}$$

where $\lambda_{\text{perf}}, \lambda_{\text{cost}} > 0$ are hyperparameters that control the performance-efficiency trade-off.

Table 7: Performance comparison on Llama1-13B across seven Common Sense Reasoning benchmarks for different pruning methods at various sparsity levels (12.5%-50%). **Boldface** indicates the best performance, and the Underline means the second-order performance. Buddy consistently outperforms baseline methods, achieving the highest average performance across all pruning ratios.

| Method | Pruning | RM Blocks | OBQA | PIQA | BoolQ | SIQA | Hellaswag | ARC-E | ARC-C | Winogrande | Avg. |
|---|---|---|---|---|---|---|---|---|---|---|---|
| Dense w/o | - | 0(0.0%) | 44.80 | 80.14 | 77.92 | 46.72 | 79.06 | 77.36 | 47.61 | 72.69 | 65.79 |
| Shortened Llama | static | 5(12.5%) | 43.00 | 79.60 | 67.43 | 47.75 | 77.38 | **76.85** | 48.46 | 71.51 | 64.00 |
| ShortGPT | static | 5(12.5%) | **44.80** | 79.54 | 73.82 | 48.67 | **78.04** | 76.47 | 48.46 | 72.85 | 65.33 |
| SLEB | static | 5(12.5%) | 43.80 | 78.56 | 74.31 | 46.47 | 76.73 | 74.92 | 43.94 | 68.90 | 63.45 |
| PuDDing | dynamic | 5(12.5%) | 42.60 | 78.45 | 67.77 | 45.45 | 76.11 | 73.36 | 43.94 | 70.24 | 62.24 |
| Buddy | dynamic | 5(12.5%) | 44.00 | **80.03** | **76.33** | **48.82** | 77.54 | 76.64 | **49.06** | **72.93** | **65.67** |
| Shortened Llama | static | 10(25%) | 40.20 | **77.75** | 58.01 | 47.65 | 73.57 | **72.98** | 44.37 | 69.06 | 60.45 |
| ShortGPT | static | 10(25%) | 42.00 | 77.58 | 54.62 | **48.31** | **74.30** | 72.22 | 45.31 | 71.27 | 60.70 |
| SLEB | static | 10(25%) | 42.00 | 77.31 | 73.06 | 46.52 | 72.97 | 72.10 | 42.83 | 64.09 | 61.36 |
| PuDDing | dynamic | 10(25%) | 38.20 | 74.43 | 69.14 | 42.22 | 69.30 | 69.95 | 38.74 | 63.06 | 58.13 |
| Buddy | dynamic | 10(25%) | **43.40** | 77.37 | **77.68** | 46.78 | 74.27 | 72.26 | **46.16** | **71.51** | **63.68** |
| Shortened Llama | static | 15(37.5%) | 38.20 | 72.80 | 49.88 | **46.11** | 68.82 | 66.37 | 40.70 | 68.03 | 56.36 |
| ShortGPT | static | 15(37.5%) | 37.40 | 72.03 | 66.97 | 45.91 | **70.12** | 65.99 | **42.06** | 68.82 | 58.66 |
| SLEB | static | 15(37.5%) | 36.40 | **73.12** | 69.88 | 43.96 | 65.51 | **66.58** | 34.64 | 61.80 | 56.49 |
| PuDDing | dynamic | 15(37.5%) | 34.00 | 69.64 | 65.84 | 38.38 | 60.13 | 60.73 | 31.66 | 58.80 | 52.40 |
| Buddy | dynamic | 15(37.5%) | **39.00** | 72.69 | **79.39** | 45.60 | 68.55 | 63.26 | 40.36 | **69.61** | **59.81** |
| Shortened Llama | static | 20(50%) | 32.40 | **69.37** | 44.13 | 42.12 | 56.01 | 57.79 | 31.83 | 59.83 | 49.18 |
| ShortGPT | static | 20(50%) | 33.80 | 66.16 | 62.32 | **42.63** | **60.04** | 54.17 | **35.75** | **64.64** | **52.44** |
| SLEB | static | 20(50%) | 34.00 | 68.99 | 61.31 | 41.30 | 54.73 | **58.00** | 30.89 | 56.43 | 50.71 |
| PuDDing | dynamic | 20(50%) | 30.60 | 64.91 | 61.99 | 37.72 | 50.38 | 48.48 | 25.68 | 55.72 | 46.94 |
| Buddy | dynamic | 20(50%) | **34.40** | 66.43 | 63.21 | 41.30 | 59.67 | 52.02 | 33.79 | 64.01 | 51.85 |

**Group-Relative Advantage.** For each training example $(x, y)$, we sample a group of $M$ actions $k^{(j)} \sim \pi_\theta(\cdot \mid G)$, $j = 1, \ldots, M$, and compute their rewards $r^{(j)} = r(k^{(j)})$. To reduce variance, we normalize rewards within each group:

$$\mu = \frac{1}{M} \sum_{j=1}^{M} r^{(j)}, \quad \sigma = \sqrt{\frac{1}{M} \sum_{j=1}^{M} (r^{(j)} - \mu)^2}, \quad A^{(j)} = \frac{r^{(j)} - \mu}{\sigma + \varepsilon}, \tag{17}$$

where $\varepsilon > 0$ ensures numerical stability.

**Optimization Objective.** GRPO maximizes the advantage-weighted log-likelihood with entropy regularization:

$$\mathcal{L}_{\text{GRPO}}(\theta) = -\frac{1}{M} \sum_{j=1}^{M} A^{(j)} \log \pi_\theta(k^{(j)} \mid G) - \beta \mathcal{H}(\pi_\theta(\cdot \mid G)), \tag{18}$$

where $\beta \geq 0$ encourages exploration and $\mathcal{H}(\cdot)$ is the categorical entropy. Only the Budget Predictor parameters $\theta$ receive gradients; the language model and Decision Module remain frozen. This directly learns a discrete layer-allocation policy that maximizes task quality per unit of compute.

## A.5 MORE EXPERIMENT RESULTS

### A.5.1 RESULTS ON LLAMA1-13B

We also conduct experiments on Llama1-13B with static baselines. Across seven common-sense reasoning benchmarks and four sparsity settings, **Buddy** exhibits strong average performance and robust degradation as pruning deepens from Table 9. At $50\%$ sparsity (removing 16 blocks), Buddy attains an average of $\mathbf{51.85}$, ranking second and trailing the best baseline (ShortGPT, $52.44$) by only $0.39$ points. For other sparsities, Buddy delivers the best average in every case: $\mathbf{65.67}$ at $12.5\%$ ($+1.67$ over the best baseline), $\mathbf{63.68}$ at $25\%$ ($+2.32$), and $\mathbf{59.81}$ at $37.5\%$ ($+1.15$). Relative to the dense (unpruned) model average of $64.36$, Buddy preserves $99.8\%$, $96.8\%$, $90.9\%$, and $78.8\%$ of accuracy at $12.5\%$, $25\%$, $37.5\%$, and $50\%$ sparsity, respectively.

### A.5.2 RESULTS ON QWEN2.5-7B-INSTRUCT

We further evaluate the performance on Qwen2.5-7B-Instruct, as Table 8 illustrates. Across all seven common-sense reasoning benchmarks, Buddy consistently outperforms existing static and

dynamic baselines, achieving the highest average accuracy at every sparsity level. At 12.5% sparsity (removing 4 blocks), Buddy reaches an average of **65.63**, slightly surpassing the best static baseline (ShortGPT, 65.49). As sparsity deepens to 28.6%, Buddy still exhibits the strongest average of **57.22**, outperforming the closest competitor (SLEB, 56.72) by 0.5 points. At 42.9% sparsity, Buddy obtains an average of **50.30**, slightly higher than SLEB (50.09) and showing a notably slower degradation than other dynamic methods. At 57.1% sparsity, Buddy maintains the top average of **45.11**, exceeding the strongest baseline (SLEB, 44.96). Relative to the dense model average of 69.42, Buddy preserves 94.5%, 82.4%, 72.5%, and 65.0% of accuracy at 14.3%, 28.6%, 42.9%, and 57.1% sparsity, respectively.

Table 8: Performance comparison on Qwen2.5-7B-Instruct across seven Common Sense Reasoning benchmarks for different pruning methods at various sparsity levels (14.3%-57.1%). **Boldface** indicates the best performance, and the Underline means the second-order performance. Buddy consistently outperforms baseline methods, achieving the highest average performance across all pruning ratios.

| Method | Pruning | RM Blocks | OBQA | PIQA | BoolQ | SIQA | Hellaswag | ARC-E | ARC-C | Winogrande | Avg. |
|---|---|---|---|---|---|---|---|---|---|---|---|
| Dense w/o | - | 0(0.0%) | 48.80 | 80.30 | 86.33 | 51.59 | 80.48 | 81.94 | 54.95 | 70.96 | 69.42 |
| Shortened Llama | static | 4(14.3%) | 44.80 | **80.36** | 78.75 | 48.26 | 73.22 | 77.86 | 48.63 | 65.19 | 64.63 |
| ShortGPT | static | 4(14.3%) | 44.00 | 77.42 | 82.39 | **50.61** | 72.95 | 79.25 | 50.51 | **66.77** | 65.49 |
| SLEB | static | 4(14.3%) | **45.20** | 79.11 | 76.24 | 48.57 | **73.25** | 77.90 | 50.34 | 62.90 | 64.19 |
| PuDDing | dynamic | 4(14.3%) | 40.60 | 75.84 | 71.01 | 45.65 | 69.30 | 72.56 | 46.67 | 59.04 | 60.08 |
| FiRST | dynamic | 4(14.3%) | 33.40 | 55.44 | 68.65 | 37.97 | 44.28 | 49.16 | 36.18 | 51.14 | 47.03 |
| Buddy | dynamic | 4(14.3%) | 43.40 | 77.26 | **84.50** | 48.93 | 73.20 | **79.88** | **51.19** | 66.69 | **65.63** |
| Shortened Llama | static | 8(28.6%) | 39.80 | 74.48 | 65.14 | 45.29 | 62.44 | 67.42 | 40.70 | 57.77 | 56.63 |
| ShortGPT | static | 8(28.6%) | 41.00 | 73.72 | 56.94 | 43.96 | 62.43 | 70.08 | 41.13 | 59.83 | 56.14 |
| SLEB | static | 8(28.6%) | **41.60** | **75.95** | 55.47 | 43.71 | **63.86** | **74.24** | **42.58** | 56.35 | 56.72 |
| PuDDing | dynamic | 8(28.6%) | 32.20 | 70.89 | 56.02 | 40.23 | 53.64 | 65.11 | 35.41 | 54.85 | 51.04 |
| FiRST | dynamic | 8(28.6%) | 26.00 | 52.34 | 55.96 | 36.18 | 35.43 | 39.52 | 28.07 | 49.01 | 40.32 |
| Buddy | dynamic | 8(28.6%) | 36.40 | 72.52 | **72.78** | **46.42** | 59.92 | 68.90 | 40.02 | **60.77** | **57.22** |
| Shortened Llama | static | 12(42.9%) | 33.20 | **71.98** | 46.61 | 41.91 | **51.90** | **65.36** | 33.02 | 50.91 | 49.36 |
| ShortGPT | static | 12(42.9%) | **37.00** | 67.36 | 57.68 | 41.56 | 48.00 | 59.26 | 32.42 | 52.41 | 49.46 |
| SLEB | static | 12(42.9%) | 35.60 | 70.35 | 51.16 | 41.91 | 51.23 | 64.56 | 33.02 | 52.88 | 50.09 |
| PuDDing | dynamic | 12(42.9%) | 32.00 | 60.77 | 50.80 | 36.54 | 36.54 | 46.72 | 26.19 | 48.78 | 42.29 |
| FiRST | dynamic | 12(42.9%) | 31.80 | 52.77 | 57.61 | 36.13 | 38.32 | 41.08 | 29.52 | 50.12 | 42.17 |
| Buddy | dynamic | 12(42.9%) | 33.20 | 67.03 | 60.70 | **41.97** | 49.25 | 60.27 | **33.36** | **56.59** | **50.30** |
| Shortened Llama | static | 16(57.1%) | 29.20 | 63.06 | 57.49 | 37.10 | 37.34 | 49.16 | 26.11 | 50.91 | 43.80 |
| ShortGPT | static | 16(57.1%) | 27.20 | 57.13 | 60.76 | 36.95 | 32.13 | 36.24 | 24.91 | 49.41 | 40.59 |
| SLEB | static | 16(57.1%) | 30.60 | **66.49** | 45.38 | **40.23** | **40.50** | **56.02** | 27.39 | **53.12** | 44.96 |
| PuDDing | dynamic | 16(57.1%) | 27.80 | 57.13 | 51.07 | 34.85 | 30.35 | 34.85 | 23.63 | 51.93 | 38.95 |
| FiRST | dynamic | 16(57.1%) | 27.00 | 51.41 | **62.17** | 34.24 | 26.67 | 25.29 | 24.57 | 50.20 | 37.69 |
| Buddy | dynamic | 16(57.1%) | **31.80** | 63.98 | 59.08 | 37.21 | 38.15 | 50.38 | **27.99** | 52.33 | **45.11** |

### A.5.3 RESULTS ON LLAMA3-8B

We further evaluate Buddy on Llama3-8B, as reported in Table 9. Across all four pruning ratios (12.5%–50%), Buddy consistently attains the highest average accuracy among both static and dynamic baselines. Relative to the dense model average of 67.44, Buddy preserves approximately 99.9%, 90.8%, 80.7%, and 71.3% of the original accuracy at 12.5%, 25%, 37.5%, and 50% sparsity, respectively, demonstrating a notably slower performance degradation under increasing pruning ratios.

### A.5.4 RESULTS ON OTHER BENCHMARKS

We further evaluate the fine-tuned Llama2-7B and Qwen2.5-7B-Instruct models on a more challenging reasoning benchmark, GSM8K (Cobbe et al., 2021). Due to the more aggressive pruning ratios, all methods suffer substantial performance degradation on this dataset. Therefore, we focus on sparsity levels of 12.5% and 25%, and report the results in Table 10. As shown in the Table 10, the performance of all pruned models is noticeably lower than their dense counterparts at both sparsity levels. Nonetheless, Buddy achieves consistently better overall performance than the baselines. The improvement is particularly significant on Qwen2.5-7B-Instruct: at 12.5% sparsity, Buddy attains 51.10 accuracy, considerably outperforming the second-best SLEB (37.15); at 25% sparsity, Buddy still leads by a large margin (14.63 vs. 6.60 for ShortGPT). These results demonstrate that

Table 9: Performance comparison on Llama3-8B across seven Common Sense Reasoning benchmarks for different pruning methods at various sparsity levels (12.5%-50%). **Boldface** indicates the best performance, and the Underline means the second-order performance. Buddy consistently outperforms baseline methods, achieving the highest average performance across all pruning ratios.

| Method | Pruning | RM Blocks | OBQA | PIQA | BoolQ | SIQA | Hellaswag | ARC-E | ARC-C | Winogrande | Avg. |
|---|---|---|---|---|---|---|---|---|---|---|---|
| Dense w/o | - | 0(0.0%) | 44.80 | 80.85 | 80.98 | 46.98 | 79.17 | 80.09 | 53.24 | 73.40 | 67.44 |
| Shortened Llama | static | 4(12.5%) | 43.40 | 79.27 | 73.06 | 47.34 | 75.32 | 76.85 | 48.04 | 66.77 | 63.76 |
| ShortGPT | static | 4(12.5%) | 44.20 | 79.76 | 79.48 | 48.16 | **76.99** | **79.88** | 51.02 | 73.72 | 66.65 |
| SLEB | static | 4(12.5%) | 44.20 | **80.47** | 75.14 | **49.28** | 76.69 | **79.88** | 50.77 | 72.22 | 66.08 |
| PuDDing | dynamic | 4(12.5%) | 38.80 | 76.82 | 68.53 | 44.83 | 72.93 | 74.12 | 44.45 | 71.98 | 61.56 |
| FiRST | dynamic | 4(12.5%) | 31.60 | 52.88 | 63.58 | 37.26 | 42.01 | 42.34 | 30.80 | 52.57 | 44.13 |
| Buddy | dynamic | 4(12.5%) | **44.80** | 79.71 | **83.24** | 48.41 | 76.25 | 78.96 | **53.33** | **74.51** | **67.40** |
| Shortened Llama | static | 8(25%) | 39.20 | **77.64** | 58.29 | 44.88 | **69.26** | **72.01** | 41.47 | 58.33 | 57.63 |
| ShortGPT | static | 8(25%) | 39.00 | 73.67 | 64.25 | 44.93 | 69.23 | 70.96 | **47.44** | **71.74** | 60.15 |
| SLEB | static | 8(25%) | **39.40** | **77.64** | 55.32 | 44.58 | 69.17 | 71.68 | 41.81 | 58.56 | 57.27 |
| PuDDing | dynamic | 8(25%) | 31.40 | 71.06 | 62.54 | 41.04 | 57.65 | 62.04 | 34.22 | 59.59 | 52.44 |
| FiRST | dynamic | 8(25%) | 33.00 | 53.44 | 64.56 | 38.02 | 46.50 | 45.08 | 31.57 | 53.83 | 45.99 |
| Buddy | dynamic | 8(25%) | 39.00 | 74.05 | **75.26** | **47.19** | 67.29 | 70.75 | 45.99 | 70.40 | **61.24** |
| Shortened Llama | static | 12(37.5%) | 35.00 | **72.69** | 61.80 | 42.68 | **59.37** | 63.76 | 33.45 | 57.54 | 53.29 |
| ShortGPT | static | 12(37.5%) | 31.60 | 68.93 | **70.24** | **43.04** | 58.97 | 56.14 | **38.40** | **67.56** | 54.36 |
| SLEB | static | 12(37.5%) | **35.80** | 72.25 | 60.52 | 42.43 | 58.78 | **65.07** | 35.92 | 56.04 | 53.35 |
| PuDDing | dynamic | 12(37.5%) | 27.00 | 64.64 | 61.93 | 38.38 | 43.13 | 47.31 | 30.46 | 57.85 | 46.34 |
| FiRST | dynamic | 12(37.5%) | 33.60 | 55.71 | 64.46 | 37.87 | 45.00 | 43.60 | 31.66 | 52.64 | 45.57 |
| Buddy | dynamic | 12(37.5%) | 34.20 | 69.97 | 68.96 | 42.32 | 58.76 | 63.05 | 37.03 | 60.93 | **54.40** |
| Shortened Llama | static | 16(50%) | 28.20 | 66.00 | 55.29 | 38.95 | 43.89 | 54.76 | 28.33 | 53.28 | 46.08 |
| ShortGPT | static | 16(50%) | 28.40 | 65.29 | **65.20** | 40.99 | **48.16** | 46.51 | 29.01 | **60.69** | 48.03 |
| SLEB | static | 16(50%) | 30.20 | **68.50** | 53.00 | **41.86** | 45.04 | **56.06** | 27.47 | 52.57 | 46.84 |
| PuDDing | dynamic | 16(50%) | 23.40 | 58.60 | 52.97 | 36.08 | 31.92 | 34.76 | 23.72 | 48.86 | 38.79 |
| FiRST | dynamic | 16(50%) | **32.40** | 54.13 | 62.60 | 37.77 | 42.80 | 42.17 | **30.20** | 52.80 | 44.36 |
| Buddy | dynamic | 16(50%) | 30.00 | 67.68 | 61.68 | 39.61 | 47.79 | 53.83 | 30.12 | 54.14 | **48.11** |

our Buddy framework remains effective even on more difficult benchmarks and under aggressive pruning regimes.

Table 10: Performance comparison on Llama2-7B and Qwen2.5-7B-Instruct on GSM8K benchmarks for different pruning methods at various sparsity levels (12.5%-25%). **Boldface** indicates the best performance, and the Underline means the second-order performance. Buddy consistently outperforms baseline methods, achieving the highest average performance across all pruning ratios.

| Method | Pruning method | rm blocks (ratio) | Llama2-7B | Qwen2.5-7B-Instruct |
|---|---|---|---|---|
| Dense w/o | - | 0(0.0%) | 14.03 | 82.87 |
| Shortened Llama | static | 4(12.5%) | 5.16 | 27.22 |
| ShortGPT | static | 4(12.5%) | **8.11** | 35.94 |
| SLEB | static | 4(12.5%) | 3.64 | 37.15 |
| PuDDing | dynamic | 4(12.5%) | 1.74 | 0.99 |
| FiRST | dynamic | 4(12.5%) | 0.91 | 27.98 |
| Buddy | dynamic | 4(12.5%) | 7.81 | **51.10** |
| Shortened Llama | static | 8(25%) | 2.05 | 5.84 |
| ShortGPT | static | 8(25%) | 2.50 | 6.60 |
| SLEB | static | 8(25%) | 1.97 | 6.37 |
| PuDDing | dynamic | 8(25%) | 0.99 | 0.61 |
| FiRST | dynamic | 8(25%) | 1.21 | 2.58 |
| Buddy | dynamic | 8(25%) | **2.65** | **14.63** |

## A.5.5 SPEED ANALYSIS

We measure end-to-end throughput (tokens/s) on ALPACA and SAMSUM for both *prefill* and *decode* phases. During decoding, the model generates 128 tokens. We compare the throughput with baselines that are dynamic inference. Results show in Table 11, indicating that in the prefill stage, Buddy outperforms existing dynamic inference methods in throughput at all sparsity rates. However, existing methods, due to excessive per-layer routing or switching between different LoRAs, have limited acceleration and can even be slower than Dense methods at low sparsity rates. While in the decode stage, Buddy left behind the PuDDing method due to its fixed inference path. In summary, the Buddy method not only offers greater versatility but also delivers greater throughput gains.

Table 11: Speed analysis on the Alpaca and SamSum datasets. The speed is measured as tokens/s.

| Method | rm blocks (ratio) | Alpaca | | | | SamSum. | | | |
|---|---|---|---|---|---|---|---|---|---|
| | | Prefill | Speed up | Decode | Speed Up | Prefill | Speed Up | Decode | Speed Up |
| Dense | 0(0.0%) | 1753.16 | ×1.00 | 39.48 | ×1.00 | 4033.83 | ×1.00 | 60.70 | ×1.00 |
| PuDDing | 4(12.5%) | 1491.73 | ×0.85 | 47.70 | ×1.21 | 3255.28 | ×0.81 | 72.84 | ×1.20 |
| FiRST | 4(12.5%) | 1794.51 | ×1.02 | 33.46 | ×0.85 | 4075.96 | ×1.01 | 51.58 | ×0.85 |
| Buddy | 4(12.5%) | 1991.54 | ×1.14 | 40.03 | ×1.01 | 4035.68 | ×1.00 | 61.28 | ×1.01 |
| PuDDing | 8(25%) | 1691.74 | ×0.96 | 55.96 | ×1.42 | 3662.78 | ×0.91 | 84.29 | ×1.39 |
| FiRST | 8(25%) | 1795.22 | ×1.02 | 33.42 | ×0.85 | 3972.77 | ×0.98 | 51.05 | ×0.84 |
| Buddy | 8(25%) | 2440.14 | ×1.39 | 46.80 | ×1.19 | 4777.07 | ×1.18 | 71.91 | ×1.18 |
| PuDDing | 12(37.5%) | 1837.07 | ×1.05 | 62.82 | ×1.59 | 3951.91 | ×0.98 | 95.53 | ×1.57 |
| FiRST | 12(37.5%) | 1780.78 | ×1.02 | 33.30 | ×0.84 | 4082.99 | ×1.01 | 51.63 | ×0.85 |
| Buddy | 12(37.5%) | 2548.39 | ×1.45 | 51.49 | ×1.30 | 5476.04 | ×1.36 | 79.11 | ×1.30 |
| PuDDing | 16(50%) | 2196.42 | ×1.25 | 81.63 | ×2.07 | 4836.26 | ×1.20 | 124.85 | ×2.06 |
| FiRST | 16(50%) | 1792.25 | ×1.02 | 33.49 | ×0.85 | 4041.14 | ×1.00 | 51.20 | ×0.84 |
| Buddy | 16(50%) | 3348.18 | ×1.91 | 64.84 | ×1.64 | 6581.55 | ×1.63 | 99.12 | ×1.63 |

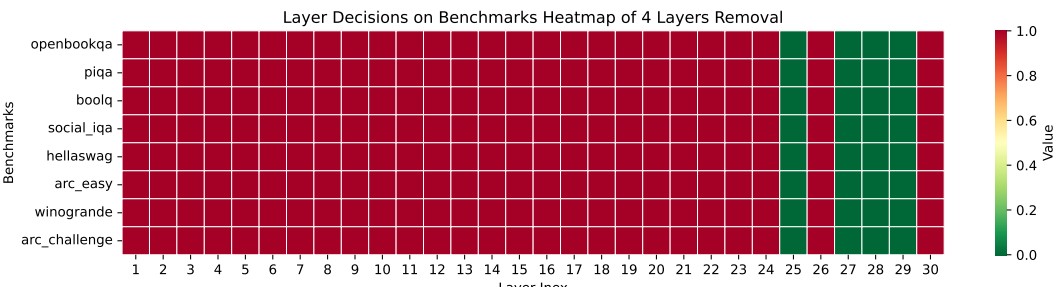

Figure 9: A visual illustration of the Buddy's average decision of each transformer block on 12.5% sparsity (4 layers removed). The color red indicates that the blocks are likely to be important, and the color green indicates that the blocks are likely to be pruned.

### A.5.6 LAYER SELECTION ANALYSIS

We analyze routing decisions across tasks and inputs under different sparsity levels; results appear in Figure 9 (12.5%), Figure 10 (25%), and Figure 11 (50%). At **low sparsity** (12.5%), layer-selection decisions are relatively stable across inputs, indicating a near-fixed path. As sparsity increases, the **importance distribution becomes less uniform**. At **25% sparsity**, layers 15, 20, 24, and 26 undergo notable shifts in importance; by **50% sparsity**, layers 9–11 exhibit the largest changes and their importance varies substantially across tasks. Moreover, at 25% sparsity the HELLASWAG task assigns different importance to layers 24 and 26, while at 50% sparsity this variability shifts to layers 9–11. These observations underscore that layer importance is *not invariant* to the sparsity regime; consequently, routing must adapt dynamically as the budget changes.

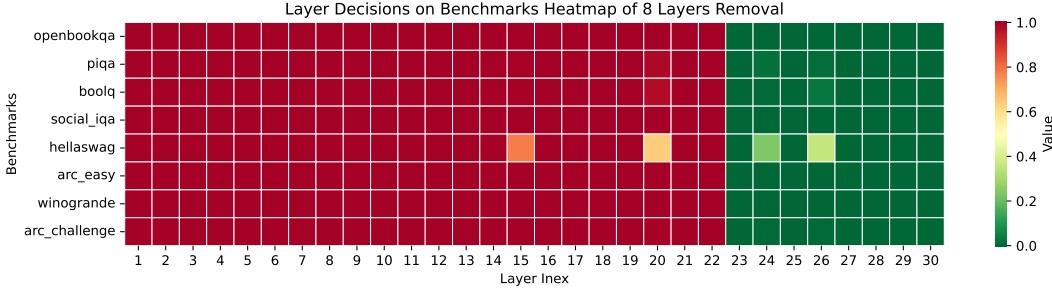

Figure 10: A visual illustration of the Buddy's average decision of each transformer block on 25% sparsity (8 layers removed). The color red indicates that the blocks are likely to be important, and the color green indicates that the blocks are likely to be pruned.

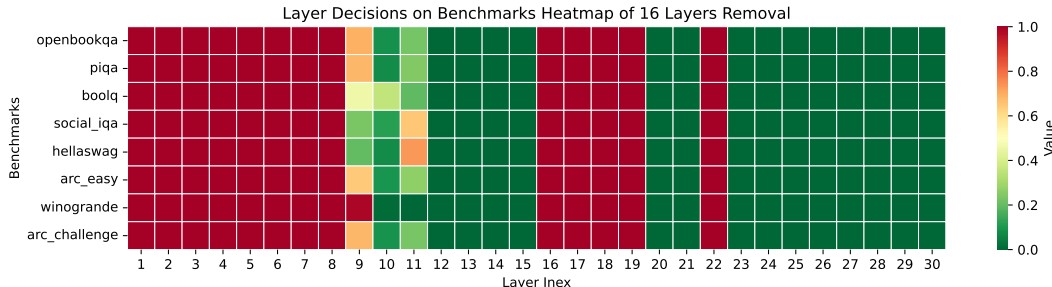

Figure 11: A visual illustration of the Buddy's average decision of each transformer block on 50% sparsity (16 layers removed). The color red indicates that the blocks are likely to be important, and the color green indicates that the blocks are likely to be pruned.

### A.5.7 ABLATION ON LAYER SELECTION FOR FEATURE EXTRACTION

We conduct an ablation study to investigate how the placement of the Decision Module, i.e., extracting contextual information from different transformer layers, affects routing quality. Specifically, we use the KV cache from layers 1 to 4 as the context input on Llama2-7B, and compare the average performance on the CommonSense benchmark, as reported in Table 12. From the results, we observe that at low sparsity levels (12.5% and 25%), using the first-layer KV cache achieves the best performance, which validates our original design choice. At higher sparsity levels, the third-layer KV cache yields slightly better results. Overall, choosing the first-layer KV cache as the context-extraction layer is a strong and robust default. When more layers are pruned, however, leveraging a slightly deeper layer as the context-extraction layer can provide richer contextual representations for routing decisions, and thus further improve performance under high sparsity.

Table 12: Ablation on the Start Layer. Average accuracy (%) is reported.

| Start layer | RM Blocks | | | |
|---|---|---|---|---|
| | 4 (12.5%) | 8 (25%) | 12 (37.5%) | 16 (50%) |
| 1 | **62.71** | **57.65** | 51.54 | 43.81 |
| 2 | 61.76 | 57.07 | 51.91 | 44.16 |
| 3 | 60.61 | 56.37 | **52.36** | **47.80** |
| 4 | 60.92 | 56.89 | 50.91 | 46.95 |

### A.5.8 BUDGET PREDICTOR TRAINING

We train the **Budget Predictor** with GRPO while *freezing* the LLM backbone and the Decision Module. Training uses the ALPACA dataset. We set the group size to $G = 4$, learning rate to $1 \times 10^{-5}$, and optimize with Adam at batch size 2 for 10,000 steps. Unless otherwise noted, we adopt the reward from Section A.4.2 with coefficients $\lambda_{\text{perf}} = 5.0$ and $\lambda_{\text{cost}} = 0.05$. Figure 12 reports the training curves of training loss, training reward, training CE, and training entropy, showing a steady decrease in $\mathcal{L}_{\text{GRPO}}$ and a concurrent increase in reward, indicating that GRPO effectively learns a stable, discrete budget policy under our setup.

### A.5.9 BUDGET PREDICTOR EVALUATION

We conduct a quantitative study on the COMMONSENSE benchmarks, evaluating *without* a user-specified budget by using our **Budget Predictor** to choose $\hat{b}$ per instance. For each benchmark, we report the average execution ratio (*execution rate*) and the fraction of dense-model performance retained (*retention rate*) to assess effectiveness. As shown in Figure 13, the predictor selects *similar* average execution across datasets (approximately $53\%$ on average), yet retention varies by task: OBQA, SIQA, and ARC-C recover over $80\%$ of dense performance, whereas HELLASWAG and ARC-E remain below $70\%$. These results indicate that budget prediction is *task-sensitive*: some

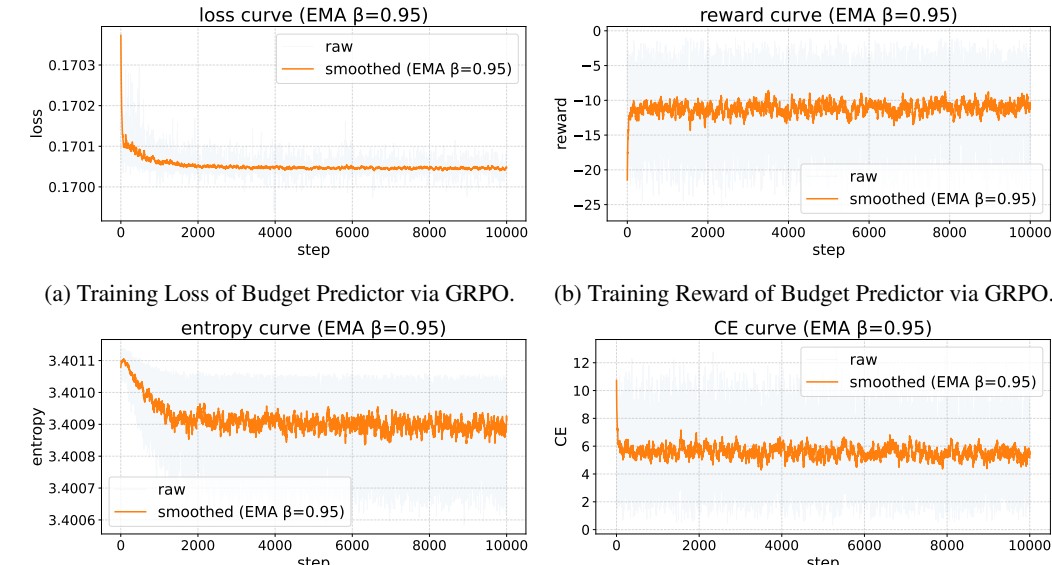

(a) Training Loss of Budget Predictor via GRPO.

(b) Training Reward of Budget Predictor via GRPO.

(c) Training Entropy of Budget Predictor via GRPO.

(d) Training CE of Budget Predictor via GRPO.

Figure 12: Training details of GRPO. Training loss, reward, CE, and entropy are reported.

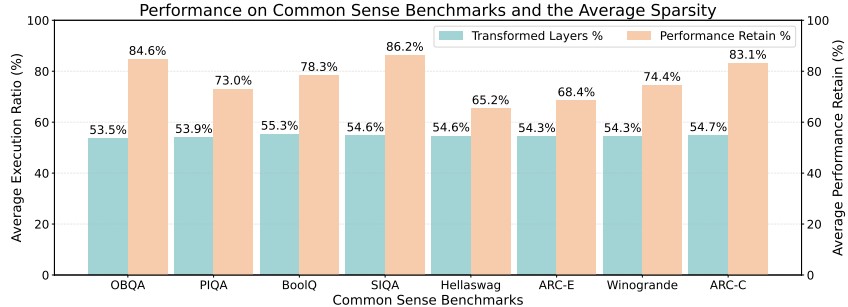

Figure 13: The average performance retention and the average execution ratio across different benchmarks, where the budgets are provided by Budget Predictor.

datasets tolerate more aggressive depth reduction at the same executed-layer ratio, while others require higher budgets to preserve accuracy.

### A.6 SOFTWARE AND HARDWARE CONFIGURATION.

Our implementation utilizes the following configurations: *PyTorch* version 2.1.2, *Transformers* library version 4.41.0, *PEFT (Parameter-Efficient Fine-Tuning)* library version 0.11.1, *CUDA* version 12.4, *GPU:* NVIDIA V100 GPU with 32GB of memory, NVIDIA A100 GPU with 80GB, *Operating System:* Ubuntu.

### A.7 LLM VERSIONS.

We provide the Hugging Face link of LLMs used in the experiment: Llama 2-7B: https://huggingface.co/meta-llama/Llama-2-7b; Llama 1-13B: https://huggingface.co/yahma/llama-13b-hf. Llama 3-8B: https://huggingface.co/meta-llama/Meta-Llama-3.1-8B. Qwen2.5-7B: https://huggingface.co/Qwen/Qwen2.5-7B-Instruct.

## A.8 LIMITATIONS AND FUTURE WORK

Because our method updates the execution path during inference to preserve performance, *all* Transformer blocks remain resident in GPU memory; thus, VRAM usage does not decrease. Besides, for batched inputs, different sequences may select different paths, which can induce KV-cache misses at skipped layers. In this paper, we use zero-filled vectors to indicate "non-executed" states; in future work we plan to develop memory-aware routing and more efficient KV-cache strategies (e.g., selective cache sharing/compaction, lightweight block swapping, and batch-level path grouping) to better support adaptive inference at scale.

