# OpenReview forum: "BUDDY: BUdget-Driven DYnamic Depth Routing for Adaptive Large Language Models Inference"
_ICLR.cc/2026/Conference — Submitted to ICLR 2026_

### Official Review · Reviewer_iBu1 · 2025-10-22

**Soundness:** 2
**Presentation:** 3
**Contribution:** 2
**Rating:** 4
**Confidence:** 4

**Summary:**

This paper proposes Buddy, a budget-driven adaptive inference framework with: (1) a Decision Module for input-dependent dynamic layer selection, (2) a Dynamic KV cache reuse and context updating mechanism, (3) a Budget Predictor for automatic cost determination.

**Strengths:**

1. Novel decode-adaptive mechanism: The framework dynamically adjusts layer selection during the decode phase (not just prefill), which represents an interesting and innovative approach to inference optimization.
2. Automatic Budget control: Automatic control over executed layers provides benefits for real-world deployment needs.
3. KV cache reuse: Leveraging the first-layer KV cache for global context is a low-overhead solution.

**Weaknesses:**

1. The experiments conducted on Llama1/Llama2, which are a bit old， Newer models (e.g., Llama3, Qwen), or thinking models (e.g., o1, DeepSeek-R1) present more challenging scenarios and would provide more relevant validation for current deployment needs.
2. The gain is a bit marginal. ShortGPT actually outperforms Buddy at 12.5% sparsity (63.16 vs 62.63), and Buddy shows only marginal improvements at higher sparsity levels.
3. Cost analysis: What is the memory overhead of maintaining the KV cache for routing decisions? Detailed latency comparisons between static and this dynamic approach are missing.
4. Only common sense reasoning benchmarks are reported. These relatively simple tasks may not effectively demonstrate the benefits of dynamic routing. (like in Figure 4, it seems all the tasks tend to prune the later layer) I would wonder if this approach would benefit more on tasks requiring multi-step reasoning or long-context understanding from decode-adaptive mechanisms.

**Questions:**

1. The Decision Module training process lacks details: Which dataset was used for training? How many samples were required? How was the ground truth routing path obtained?
2. Ablations comparing different KV cache selection strategies: Why specifically use the first layer's KV cache? What about other layers?
3. In real deployment, contexts can shift completely (e.g., switching from technical discussion to casual chat, or processing unrelated prompts in the same session).  Would KV cache from irrelevant prior contexts influence layer selection?

---

> ### Author Response · Authors · 2025-11-26
>
> We sincerely thank you for your detailed feedback and constructive suggestions. We have carefully considered every
> comment and conducted additional experiments and analyses where necessary. Below, we address each comment individually
> and provide clarifications, updates, and results that demonstrate how the concerns have been addressed.
>
> **Comment:** "*The experiments conducted on Llama1/Llama2, which are a bit old， Newer models (e.g., Llama3, Qwen), or
> thinking models (e.g., o1, DeepSeek-R1) present more challenging scenarios and would provide more relevant validation
> for current deployment needs.*"
>
> **Response**: Thank you for your valuable comment. We agree that validating on newer models strengthens practical
> relevance. Accordingly, we extended our evaluation to Llama-3-8B and Qwen2.5-7B-Instruct, two of the most widely adopted
> open-source modern LLMs. As shown in the following tables and the revised Appendix. Buddy consistently surpasses
> baseline pruning approaches on both architectures, confirming that our method generalizes well beyond earlier LLaMA
> models.

---

> ### Author Response · Authors · 2025-11-26
>
> Results on Qwen2.5-7B-Instruct
>
> | Method          | pruning method | rm blocks (ratio) | obqa         | piqa         | boolq        | siqa         | hellaswag    | arc-e        | arc-c        | winogrande   | avg.         |
> |-----------------|----------------|-------------------|--------------|--------------|--------------|--------------|--------------|--------------|--------------|--------------|--------------|
> | Dense w/o       | -              | 0(0.0%)           | 48.80        | 80.30        | 86.33        | 51.59        | 80.48        | 81.94        | 54.95        | 70.96        | 69.42        |
> | Shortened Llama | static         | 4(14.3%)          | 44.80 | **80.36**    | 78.75        | 48.26        | 73.22 | 77.86        | 48.63        | 65.19        | 64.63        |
> | ShortGPT        | static         | 4(14.3%)          | 44.00        | 77.42        | 82.39 | **50.61**    | 72.95        | 79.25 | 50.51 | **66.77**    | 65.49 |
> | SLEB            | static         | 4(14.3%)          | **45.20**    | 79.11 | 76.24        | 48.57        | **73.25**    | 77.90        | 50.34        | 62.90        | 64.19        |
> | PuDDing         | dynamic        | 4(14.3%)          | 40.60        | 75.84        | 71.01        | 45.65        | 69.30        | 72.56        | 46.67        | 59.04        | 60.08        |
> | FiRST           | dynamic        | 4(14.3%)          | 33.40        | 55.44        | 68.65        | 37.97        | 44.28        | 49.16        | 36.18        | 51.14        | 47.03        |
> | Buddy           | dynamic        | 4(14.3%)          | 43.40        | 77.26        | **84.50**    | 48.93 | 73.20        | **79.88**    | **51.19**    | 66.69 | **65.63**    |
> | Shortened Llama | static         | 8(28.6%)          | 39.80        | 74.48 | 65.14 | 45.29 | 62.44 | 67.42        | 40.70        | 57.77        | 56.63        |
> | ShortGPT        | static         | 8(28.6%)          | 41.00 | 73.72        | 56.94        | 43.96        | 62.43        | 70.08 | 41.13 | 59.83 | 56.14        |
> | SLEB            | static         | 8(28.6%)          | **41.60**    | **75.95**    | 55.47        | 43.71        | **63.86**    | **74.24**    | **42.58**    | 56.35        | 56.72 |
> | PuDDing         | dynamic        | 8(28.6%)          | 32.20        | 70.89        | 56.02        | 40.23        | 53.64        | 65.11        | 35.41        | 54.85        | 51.04        |
> | FiRST           | dynamic        | 8(28.6%)          | 26.00        | 52.34        | 55.96        | 36.18        | 35.43        | 39.52        | 28.07        | 49.01        | 40.32        |
> | Buddy           | dynamic        | 8(28.6%)          | 36.40        | 72.52        | **72.78**    | **46.42**    | 59.92        | 68.90        | 40.02        | **60.77**    | **57.22**    |
> | Shortened Llama | static         | 12(42.9%)         | 33.20        | **71.98**    | 46.61        | 41.91 | **51.90**    | **65.36**    | 33.02 | 50.91        | 49.36        |
> | ShortGPT        | static         | 12(42.9%)         | **37.00**    | 67.36        | 57.68 | 41.56        | 48.00        | 59.26        | 32.42        | 52.41        | 49.46        |
> | SLEB            | static         | 12(42.9%)         | 35.60 | 70.35 | 51.16        | 41.91 | 51.23 | 64.56 | 33.02 | 52.88 | 50.09 |
> | PuDDing         | dynamic        | 12(42.9%)         | 32.00        | 60.77        | 50.80        | 36.54        | 36.54        | 46.72        | 26.19        | 48.78        | 42.29        |
> | FiRST           | dynamic        | 12(42.9%)         | 31.80        | 52.77        | 57.61        | 36.13        | 38.32        | 41.08        | 29.52        | 50.12        | 42.17        |
> | Buddy           | dynamic        | 12(42.9%)         | 33.20        | 67.03        | **60.70**    | **41.97**    | 49.25        | 60.27        | **33.36**    | **56.59**    | **50.30**    |
> | Shortened Llama | static         | 16(57.1%)         | 29.20        | 63.06        | 57.49        | 37.10        | 37.34        | 49.16        | 26.11        | 50.91        | 43.80        |
> | ShortGPT        | static         | 16(57.1%)         | 27.20        | 57.13        | 60.76 | 36.95        | 32.13        | 36.24        | 24.91        | 49.41        | 40.59        |
> | SLEB            | static         | 16(57.1%)         | 30.60 | **66.49**    | 45.38        | **40.23**    | **40.50**    | **56.02**    | 27.39 | **53.12**    | 44.96 |
> | PuDDing         | dynamic        | 16(57.1%)         | 27.80        | 57.13        | 51.07        | 34.85        | 30.35        | 34.85        | 23.63        | 51.93        | 38.95        |
> | FiRST           | dynamic        | 16(57.1%)         | 27.00        | 51.41        | **62.17**    | 34.24        | 26.67        | 25.29        | 24.57        | 50.20        | 37.69        |
> | Buddy           | dynamic        | 16(57.1%)         | **31.80**    | 63.98 | 59.08        | 37.21 | 38.15 | 50.38 | **27.99**    | 52.33 | **45.11**    |

---

> ### Author Response · Authors · 2025-11-26
>
> Results on Llama3-8B
>
> | Method          | pruning method | rm blocks (ratio) | obqa         | piqa         | boolq        | siqa         | hellaswag    | arc-e        | arc-c        | winogrande   | avg.         |
> |-----------------|----------------|-------------------|--------------|--------------|--------------|--------------|--------------|--------------|--------------|--------------|--------------|
> | Shortened Llama | static         | 4(12.5%)          | 43.40        | 79.27        | 73.06        | 47.34        | 75.32        | 76.85        | 48.04        | 66.77        | 63.76        |
> | ShortGPT        | static         | 4(12.5%)          | 44.20 | 79.76 | 79.48 | 48.16        | **76.99**    | **79.88**    | 51.02 | 73.72 | 66.65 |
> | SLEB            | static         | 4(12.5%)          | 44.20 | **80.47**    | 75.14        | **49.28**    | 76.69 | **79.88**    | 50.77        | 72.22        | 66.08        |
> | PuDDing         | dynamic        | 4(12.5%)          | 38.80        | 76.82        | 68.53        | 44.83        | 72.93        | 74.12        | 44.45        | 71.98        | 61.56        |
> | FiRST           | dynamic        | 4(12.5%)          | 31.60        | 52.88        | 63.58        | 37.26        | 42.01        | 42.34        | 30.80        | 52.57        | 44.13        |
> | Buddy           | dynamic        | 4(12.5%)          | **44.80**    | 79.71        | **83.24**    | 48.41 | 76.25        | 78.96 | **53.33**    | **74.51**    | **67.40**    |
> | Shortened Llama | static         | 8(25%)            | 39.20 | **77.64**    | 58.29        | 44.88        | **69.26**    | **72.01**    | 41.47        | 58.33        | 57.63        |
> | ShortGPT        | static         | 8(25%)            | 39.00        | 73.67        | 64.25        | 44.93 | 69.23 | 70.96        | **47.44**    | **71.74**    | 60.15 |
> | SLEB            | static         | 8(25%)            | **39.40**    | **77.64**    | 55.32        | 44.58        | 69.17        | 71.68 | 41.81        | 58.56        | 57.27        |
> | PuDDing         | dynamic        | 8(25%)            | 31.40        | 71.06        | 62.54        | 41.04        | 57.65        | 62.04        | 34.22        | 59.59        | 52.44        |
> | FiRST           | dynamic        | 8(25%)            | 33.00        | 55.33        | 64.56 | 38.02        | 46.50        | 45.08        | 31.57        | 53.83        | 45.99        |
> | Buddy           | dynamic        | 8(25%)            | 39.00        | 74.05 | **75.26**    | **47.19**    | 67.29        | 70.75        | 45.99 | 70.40 | **61.24**    |
> | Shortened Llama | static         | 12(37.5%)         | 35.00 | **72.69**    | 61.80        | 42.68 | **59.37**    | 63.76 | 33.45        | 57.54        | 53.29        |
> | ShortGPT        | static         | 12(37.5%)         | 31.60        | 68.93        | **70.24**    | **43.04**    | 58.97 | 56.14        | **38.40**    | **67.56**    | 54.36 |
> | SLEB            | static         | 12(37.5%)         | **35.80**    | 72.25 | 60.52        | 42.43        | 58.78        | **65.07**    | 35.92        | 56.04        | 53.35        |
> | PuDDing         | dynamic        | 12(37.5%)         | 27.00        | 64.64        | 61.93        | 38.38        | 43.13        | 47.31        | 30.46        | 57.85        | 46.34        |
> | FiRST           | dynamic        | 12(37.5%)         | 33.60        | 55.71        | 64.46 | 37.87        | 45.00        | 43.60        | 31.66        | 52.64        | 45.57        |
> | Buddy           | dynamic        | 12(37.5%)         | 34.20        | 69.97        | 68.96 | 42.32        | 58.76        | 63.05        | 37.03 | 60.93 | **54.40**    |
> | Shortened Llama | static         | 16(50%)           | 28.20        | 66.00        | 55.29        | 38.95        | 43.89        | 54.76 | 28.33        | 53.28        | 46.08        |
> | ShortGPT        | static         | 16(50%)           | 28.40        | 65.29        | **65.20**    | 40.99 | **48.16**    | 46.51        | 29.01        | **60.69**    | 48.03 |
> | SLEB            | static         | 16(50%)           | 30.20 | **68.50**    | 53.00        | **41.86**    | 45.04        | **56.06**    | 27.47        | 52.57        | 46.84        |
> | PuDDing         | dynamic        | 16(50%)           | 23.40        | 58.60        | 52.97        | 36.08        | 31.92        | 34.76        | 23.72        | 48.86        | 38.79        |
> | FiRST           | dynamic        | 16(50%)           | **32.40**    | 54.13        | 62.60 | 37.77        | 42.80        | 42.17        | **30.20**    | 52.80        | 44.36        |
> | Buddy           | dynamic        | 16(50%)           | 30.00        | 67.68 | 61.68        | 39.61        | 47.79 | 53.83        | 30.12 | 54.14 | **48.11**    |

---

> ### Author Response · Authors · 2025-11-26
>
> **Comment:** "The gain is a bit marginal. ShortGPT actually outperforms Buddy at 12.5% sparsity (63.16 vs 62.63), and
> Buddy shows only marginal improvements at higher sparsity levels."
>
> **Response**: Thank you for your thoughtful feedback. In our main results, we evaluate four sparsity levels; Buddy is *
> *the top-performing approach in three of them, and at 12.5% sparsity it is slightly behind the best baseline,** ranking
> second overall. This pattern suggests that Buddy is generally strong across sparsity settings, even if the improvement
> is not large at every single point. At the same time, our goal is to **introduce a budget-aware dynamic routing
> framework that can strictly respect user budgets and flexibly adapt to different compute constraints** using a single
> model, as highlighted in Table 1. Buddy accomplishes this by dynamically selecting layer execution paths according to
> the specified budget via the Decision Module. Existing static pruning and many prior dynamic methods, however, are
> typically **designed for one fixed sparsity and would require separate models** to cover multiple speed–accuracy
> trade-offs, which limits their usability when users frequently change budgets. In contrast, Buddy combines good
> empirical performance with a much more flexible and practical deployment story, which we believe is a key strength of
> the proposed framework.

---

> ### Author Response · Authors · 2025-11-26
>
> **Comment:** "*Cost analysis: What is the memory overhead of maintaining the KV cache for routing decisions? Detailed
> latency comparisons between static and this dynamic approach are missing.*"
>
> **Response**: Thank you for the thoughtful comment. We address the two aspects—memory overhead and latency:
>
> (1) Memory overhead of using the first-layer KV cache. Buddy **does not introduce any additional KV-cache memory**
> beyond what the base LLM already maintains. During standard autoregressive decoding, all Transformer layers, including
> the first, store their own KV states. Our method simply reuses the existing first-layer KV cache as an input feature to
> the Decision Module, without storing any extra copies or additional tensors. **Thus, the memory footprint of KV caching
> in Buddy is identical to that of standard LLM decoding**, and no additional VRAM is consumed for routing decisions.
>
> (2) Latency comparison with static pruning. We agree that static pruning methods enjoy an inherent speed advantage
> because they predefine a fixed computation graph after pruning, resulting in zero routing overhead at inference. In
> contrast, Buddy performs dynamic budget-aware routing, where the Decision Module evaluates the current context to select
> the execution path. **This flexibility inevitably introduces a small amount of extra computation and the latency from
> undetermined computation graph**, which is the common challenge across dynamic inference methods.
>
> However, Buddy still provides non-trivial speedups over the dense model (see Table 3), while achieving higher accuracy
> than static pruning baselines at medium and high sparsity levels. In other words, although dynamic methods cannot match
> the absolute minimal latency of fully static pruning, Buddy offers a more favorable accuracy–latency trade-off by
> adapting computation to the input rather than pruning layers permanently. Moreover, **Buddy only introduce one model to
> serve at different sparsity requirements, which can migrate the memory overhead of deploying different pruned models
> with different sparsity**.
>
> We acknowledge that dynamic routing can be further optimized at the systems level. As future work, we plan to implement
> operator-level and kernel-level fusion for routing decisions, which can reduce routing overhead and narrow the latency
> gap with static pruning, while retaining the adaptability benefits of our method.

---

> ### Author Response · Authors · 2025-11-26
>
> **Comment:** "*Only common sense reasoning benchmarks are reported. These relatively simple tasks may not effectively
> demonstrate the benefits of dynamic routing. (like in Figure 4, it seems all the tasks tend to prune the later layer) I
> would wonder if this approach would benefit more on tasks requiring multi-step reasoning or long-context understanding
> from decode-adaptive mechanisms.*"
>
> **Response**: Thank you for pointing this out. We agree that commonsense benchmarks alone may not fully reveal the
> potential of dynamic layer routing. To directly address this concern, we added a more challenging evaluation on GSM8K,
> which requires complex, multi-step numerical reasoning and is far more sensitive to layer selection than commonsense QA.
> We evaluated Buddy and all baselines on GSM8K using both Llama-2-7B and Qwen2.5-7B. The results confirm that Buddy
> consistently surpasses static pruning methods and recent dynamic baselines, especially at moderate and high sparsity
> levels. This suggests that Buddy’s dynamic routing indeed provides larger gains on reasoning-heavy tasks. These results
> have been added to the revised Appendix.
>
> | Method          | pruning method | rm blocks (ratio) | Llama2-7B   | Qwen2.5-7B-Instruct |
> |-----------------|----------------|-------------------|-------------|---------------------|
> | Dense w/o       | -              | 0(0.0%)           | 14.03       | 82.87               |
> | Shortened Llama | static         | 4(12.5%)          | 5.16        | 27.22               |
> | ShortGPT        | static         | 4(12.5%)          | **8.11**    | 35.94               |
> | SLEB            | static         | 4(12.5%)          | 3.64        | 37.15        |
> | PuDDing         | dynamic        | 4(12.5%)          | 1.74        | 0.99                |
> | FiRST           | dynamic        | 4(12.5%)          | 0.91        | 27.98               |
> | Buddy           | dynamic        | 4(12.5%)          | 7.81 | **51.10**           |
> | Shortened Llama | static         | 8(25%)            | 2.05        | 5.84                |
> | ShortGPT        | static         | 8(25%)            | 2.50 | 6.60         |
> | SLEB            | static         | 8(25%)            | 1.97        | 6.37                |
> | PuDDing         | dynamic        | 8(25%)            | 0.99        | 0.61                |
> | FiRST           | dynamic        | 8(25%)            | 1.21        | 2.58                |
> | Buddy           | dynamic        | 8(25%)            | **2.65**    | **14.63**           |

---

> ### Author Response · Authors · 2025-11-26
>
> **Question:** "*The Decision Module training process lacks details: Which dataset was used for training? How many
> samples were required? How was the ground truth routing path obtained?*"
>
> **Response**: Thank you for raising these important points. The Decision Module **is trained via supervised
> fine-tuning (SFT) together with the LoRA weights of the backbone model on the Alpaca instruction dataset.** In practice,
> we use the same SFT setup as for standard instruction tuning: the model is trained on the full Alpaca training split,
> and we minimize the cross-entropy loss for next-token prediction. The Decision Module **does not require an explicit
> “ground truth routing path.”** Instead, the routing policy is learned implicitly: we treat the layer selection as a
> latent decision that is optimized to minimize the language modeling loss. Concretely, the MLP in the Decision Module
> produces per-layer scores, **we apply Top-k gating with a straight-through estimator (STE), and gradients from the
> cross-entropy loss are back-propagated into the MLP parameters**. Thus, the Decision Module learns to select informative
> layers purely from the end-task supervision, without any manually defined or precomputed routing labels. We have
> clarified these details—dataset, training procedure, and supervision signal in the revised **Section 3.2**, **Section
> 4.1**, and **Appendix A.4.1**.

---

> ### Author Response · Authors · 2025-11-26
>
> **Question:** "*Ablations comparing different KV cache selection strategies: Why specifically use the first layer's KV
> cache? What about other layers?*"
>
> **Response**: Thank you for this insightful question. Our original motivation for using the first-layer KV cache was to
> obtain a lightweight and universally available summary of the entire context during both the prefill and decode stages.
> **Since the first layer processes every token before any deeper transformation occurs, its KV states provide a clean and
> stable global representation with minimal computational overhead**. This made it a natural starting point for our
> design.
> Following the reviewer’s suggestion, we performed an ablation study to **examine whether using KV caches from other
> early layers would affect routing quality.** Specifically, we experimented with using the KV cache **from layers 1
> through 4 as the context input** to the Decision Module. Results averaged across our benchmark suite are shown below:
>
> | Start layer | RM Blocks |           |            |           |
> |-------------|-----------|-----------|------------|-----------|
> |             | 4 (12.5%) | 8 (25%)   | 12 (37.5%) | 16 (50%)  |
> | 1           | **62.71** | **57.65** | 51.54      | 43.81     |
> | 2           | 61.76     | 57.07     | 51.91      | 44.16     |
> | 3           | 60.61     | 56.37     | **52.36**  | **47.80** |
> | 4           | 60.92     | 56.89     | 50.91      | 46.95     |
>
> From the results, we find that at low sparsity levels (12.5% and 25%), the first-layer KV cache achieves the best
> performance, confirming our original design choice. At higher sparsity levels, the third-layer KV cache yields slightly
> better results. We hypothesize that when more layers are pruned, retaining information from earlier layers provides a
> richer contextual representation for routing decisions, which conforms to the result from our analysis of layer
> selection, where the earlier layers should be retained, and middle layers are more suitable to be pruned.
>
> Overall, the first layer remains a strong and robust default choice across sparsity regimes, while deeper layers may
> offer marginal benefits under extreme pruning. We have incorporated this ablation and discussion into the revised
> submission, and we thank the reviewer again for prompting this useful analysis.

---

> ### Author Response · Authors · 2025-11-26
>
> **Question:** "*In real deployment, contexts can shift completely (e.g., switching from technical discussion to
> casual chat, or processing unrelated prompts in the same session). Would KV cache from irrelevant prior contexts
> influence layer selection?"*
>
> **Response**: Thank you for raising this important question. We agree that, in practical deployments, user contexts may
> shift abruptly (e.g., from a technical query to casual conversation). We clarify how Buddy behaves in such scenarios.
>
> Buddy does not introduce any additional persistence of historical context beyond what standard autoregressive LLMs
> already maintain. **Our method simply reuses the first-layer KV cache that is already part of the model’s decoding state
> **.
> Thus, if the underlying LLM continues to condition on past KV cache after a sudden topic shift, then **Buddy’s
> layer-selection module will likewise reflect the same context, consistent with the model’s native behavior.** In other
> words, this is not a phenomenon introduced by Buddy, but an inherent property of LLM decoding: if the base model keeps
> the KV cache, it will condition on prior content; if the application resets the KV cache (as is commonly done when
> starting a new conversation or switching contexts), Buddy will immediately adapt because its routing decisions are
> recomputed from the new KV signals at each step.
>
> We appreciate the reviewer for highlighting this scenario, as it suggests a promising direction for future improvements.
> Incorporating explicit context-boundary detection or automatic KV reset strategies could further enhance routing
> stability under abrupt topic shifts. We will explore this in future work.

---

> > ### Comment · Reviewer_iBu1 · 2025-11-26
> >
> > Thank you for the detailed response and additional experiments.
> > However, I remain concerned about practical applicability. The performance drop on GSM8K compared to dense models is unacceptable for any practical applications (static pruning methods are easier to fine-tune for recovery). Additionally, the marginal gains over static depth pruning on language tasks do not justify the added complexity and potential inability to batch inference efficiently.
> > For these reasons, I maintain my original score.

---

### Official Review · Reviewer_FF78 · 2025-10-27

**Soundness:** 3
**Presentation:** 3
**Contribution:** 2
**Rating:** 4
**Confidence:** 4

**Summary:**

This paper introduces a budget-driven and adaptive inference framework for large language models. It dynamically selects important layers during inference through a Decision Module, reuses and updates KV caches for contextual adaptation, and predicts budgets when unspecified.

**Strengths:**

Quality

1. The paper is well written and easy to follow.

Significance

2. The analysis showing that the optimal routing path can change during decoding is interesting and provides new insight into adaptive inference.

**Weaknesses:**

Originality

1. The gating-based layer skipping mechanism is not entirely novel. Similar techniques have been explored in prior works [1-4].

Quality

2. The model used in experiments is somewhat outdated, which limits the relevance of the reported results.

Clarity

3. The paper does not clearly explain how the Decision Module is trained, leaving ambiguity around the optimization and supervision of the gating mechanism.
4. The calibration (validation) set needs to be described in more detail for better clarity.

Significance

5. The experimental results are not strong enough to convincingly highlight the benefits of the proposed framework. Evaluations on more challenging or reasoning-intensive tasks could better showcase its advantages.
6. The Budget Predictor is presented as part of the framework but appears tangential to the main contribution, with minimal results shown in the main paper.

[1] DASH: Input-Aware Dynamic Layer Skipping for Efficient LLM Inferencewith Markov Decision Policies
[2] DiffSkip: Differential Layer Skipping in Large Language Models
[3] What Layers When: Learning to Skip Compute in LLMs with Residual Gates
[4] Skip Transformers: Efficient Inference throughSkip-Routing

**Questions:**

1. How is the Decision Module trained? Please clarify the optimization process for the gating mechanism.
2. The gating mechanism for layer skipping has been explored in prior works. The authors are encouraged to discuss these related works [1–4] in the related work section and include some of them as baselines in the experiments.
3. Minor typo: remove the duplicate period in “a single deployed LLM to serve heterogeneous budgets..”.

[1] DASH: Input-Aware Dynamic Layer Skipping for Efficient LLM Inferencewith Markov Decision Policies
[2] DiffSkip: Differential Layer Skipping in Large Language Models
[3] What Layers When: Learning to Skip Compute in LLMs with Residual Gates
[4] Skip Transformers: Efficient Inference throughSkip-Routing

---

> ### Author Response · Authors · 2025-11-26
>
> We sincerely thank you for your detailed feedback and constructive suggestions. We have carefully considered every comment and conducted additional experiments and analyses where necessary. Below, we address each comment individually and provide clarifications, updates, and results that demonstrate how the concerns have been addressed.
>
> **Comment:** "*The gating-based layer skipping mechanism is not entirely novel. Similar techniques have been explored in prior works [1-4].*"
>
> **Question:** "*The gating mechanism for layer skipping has been explored in prior works. The authors are encouraged to discuss these related works [1–4] in the related work section and include some of them as baselines in the experiments.*"
>
> **Response**:
> Thank you very much for your constructive comment. We fully agree that gating-based mechanisms for dynamic computation have been explored in prior work, and we appreciate the reviewers pointing us to [1–4]. We clarify below how these approaches relate to our method and what the difference is.
>
> First, the works referenced by the reviewer focus primarily on token-wise dynamic skipping, where each layer selectively computes only a subset of tokens to reduce computation. In contrast, our method targets layer-level dynamic execution, where all tokens participate in every executed layer, but the model skips entire Transformer blocks based on input-dependent importance. Thus, while both families use gating modules, their granularity and optimization objectives are fundamentally different:
>
> - **Token-level gating** (as in [1–4]) aims to reduce computation by dynamically choosing which tokens to process at each layer. These approaches are powerful and fine-grained, but the resulting computational cost can vary unpredictably, making it difficult to respect a strict user-specified budget.
> - **Layer-level gating** (our setting) deterministically selects which blocks to execute such that the number of executed layers exactly matches a given budget. This strict compute control is essential for budget-driven inference, as emphasized in Table 1 of our paper.
>
> Second, many token-wise skipping methods cannot guarantee a fixed computational cost during inference because their effective FLOPs depend on the number and distribution of “kept tokens.” In contrast, Buddy’s routing mechanism is explicitly designed to strictly satisfy user-defined budgets, which is a key motivation of our work and differentiates it from these prior approaches. Following the reviewer’s suggestion, we have updated the related work section to explicitly discuss these token-level dynamic computation methods and added them to Table 1 for a clearer comparison.
>
> Finally, we reproduced and evaluated two representative gating-based dynamic baselines—Skip-Transformer and Diff-Skip—under our unified training setup. We tuned their hyperparameters to match the target sparsity levels used in our experiments.

---

> > ### Author Response · Authors · 2025-11-26
> >
> > **Comment:** "*The model used in experiments is somewhat outdated, which limits the relevance of the reported results.*"
> >
> > **Response**: Thank you for the valuable feedback. While we initially focused on Llama-2 and Llama-1 to ensure a fair comparison with prior depth-pruning work, we fully agree that evaluating on newer architectures improves practical relevance. Therefore, we performed additional experiments on Llama-3-8B and Qwen2.5-7B-Instruct. Across both architectures, Buddy consistently achieves higher accuracy–compute trade-offs than baselines. These results further support our claim that Buddy’s design is model-independent and remains effective across modern LLM families. The updated appendix includes all experimental details.
> >
> > Results on Qwen2.5-7B-Instruct
> >
> > | Method | pruning method | rm blocks (ratio) | obqa | piqa | boolq | siqa | hellaswag | arc-e | arc-c | winogrande | avg. |
> > | --- | --- | --- | --- | --- | --- | --- | --- | --- | --- | --- | --- |
> > | Dense w/o | - | 0(0.0%) | 48.80 | 80.30 | 86.33 | 51.59 | 80.48 | 81.94 | 54.95 | 70.96 | 69.42 |
> > | Shortened Llama | static | 4(14.3%) | 44.80 | **80.36** | 78.75 | 48.26 | 73.22 | 77.86 | 48.63 | 65.19 | 64.63 |
> > | ShortGPT | static | 4(14.3%) | 44.00 | 77.42 | 82.39 | **50.61** | 72.95 | 79.25 | 50.51 | **66.77** | 65.49 |
> > | SLEB | static | 4(14.3%) | **45.20** | 79.11 | 76.24 | 48.57 | **73.25** | 77.90 | 50.34 | 62.90 | 64.19 |
> > | PuDDing | dynamic | 4(14.3%) | 40.60 | 75.84 | 71.01 | 45.65 | 69.30 | 72.56 | 46.67 | 59.04 | 60.08 |
> > | FiRST | dynamic | 4(14.3%) | 33.40 | 55.44 | 68.65 | 37.97 | 44.28 | 49.16 | 36.18 | 51.14 | 47.03 |
> > | Buddy | dynamic | 4(14.3%) | 43.40 | 77.26 | **84.50** | 48.93 | 73.20 | **79.88** | **51.19** | 66.69 | **65.63** |
> > | Shortened Llama | static | 8(28.6%) | 39.80 | 74.48 | 65.14 | 45.29 | 62.44 | 67.42 | 40.70 | 57.77 | 56.63 |
> > | ShortGPT | static | 8(28.6%) | 41.00 | 73.72 | 56.94 | 43.96 | 62.43 | 70.08 | 41.13 | 59.83 | 56.14 |
> > | SLEB | static | 8(28.6%) | **41.60** | **75.95** | 55.47 | 43.71 | **63.86** | **74.24** | **42.58** | 56.35 | 56.72 |
> > | PuDDing | dynamic | 8(28.6%) | 32.20 | 70.89 | 56.02 | 40.23 | 53.64 | 65.11 | 35.41 | 54.85 | 51.04 |
> > | FiRST | dynamic | 8(28.6%) | 26.00 | 52.34 | 55.96 | 36.18 | 35.43 | 39.52 | 28.07 | 49.01 | 40.32 |
> > | Buddy | dynamic | 8(28.6%) | 36.40 | 72.52 | **72.78** | **46.42** | 59.92 | 68.90 | 40.02 | **60.77** | **57.22** |
> > | Shortened Llama | static | 12(42.9%) | 33.20 | **71.98** | 46.61 | 41.91 | **51.90** | **65.36** | 33.02 | 50.91 | 49.36 |
> > | ShortGPT | static | 12(42.9%) | **37.00** | 67.36 | 57.68 | 41.56 | 48.00 | 59.26 | 32.42 | 52.41 | 49.46 |
> > | SLEB | static | 12(42.9%) | 35.60 | 70.35 | 51.16 | 41.91 | 51.23 | 64.56 | 33.02 | 52.88 | 50.09 |
> > | PuDDing | dynamic | 12(42.9%) | 32.00 | 60.77 | 50.80 | 36.54 | 36.54 | 46.72 | 26.19 | 48.78 | 42.29 |
> > | FiRST | dynamic | 12(42.9%) | 31.80 | 52.77 | 57.61 | 36.13 | 38.32 | 41.08 | 29.52 | 50.12 | 42.17 |
> > | Buddy | dynamic | 12(42.9%) | 33.20 | 67.03 | **60.70** | **41.97** | 49.25 | 60.27 | **33.36** | **56.59** | **50.30** |
> > | Shortened Llama | static | 16(57.1%) | 29.20 | 63.06 | 57.49 | 37.10 | 37.34 | 49.16 | 26.11 | 50.91 | 43.80 |
> > | ShortGPT | static | 16(57.1%) | 27.20 | 57.13 | 60.76 | 36.95 | 32.13 | 36.24 | 24.91 | 49.41 | 40.59 |
> > | SLEB | static | 16(57.1%) | 30.60 | **66.49** | 45.38 | **40.23** | **40.50** | **56.02** | 27.39 | **53.12** | 44.96 |
> > | PuDDing | dynamic | 16(57.1%) | 27.80 | 57.13 | 51.07 | 34.85 | 30.35 | 34.85 | 23.63 | 51.93 | 38.95 |
> > | FiRST | dynamic | 16(57.1%) | 27.00 | 51.41 | **62.17** | 34.24 | 26.67 | 25.29 | 24.57 | 50.20 | 37.69 |
> > | Buddy | dynamic | 16(57.1%) | **31.80** | 63.98 | 59.08 | 37.21 | 38.15 | 50.38 | **27.99** | 52.33 | **45.11** |

---

> ### Author Response · Authors · 2025-11-26
>
> | Method | pruning method | rm blocks (ratio) | obqa | piqa | boolq | siqa | hellaswag | arc-e | arc-c | winogrande | avg. |
> | --- | --- | --- | --- | --- | --- | --- | --- | --- | --- | --- | --- |
> | Dense w/o | - | 0(0.0%) | 44.20 | 79.11 | 77.71 | 46.06 | 76.02 | 76.30 | 46.33 | 69.14 | 64.36 |
> | Shortened Llama | static | 4(12.5%) | 40.60 | **78.02** | 71.25 | 45.29 | 72.53 | 73.27 | 42.83 | 62.90 | 60.84 |
> | ShortGPT | static | 4(12.5%) | 42.00 | 77.15 | **78.69** | **47.59** | **73.69** | 72.56 | 44.45 | **69.14** | **63.16** |
> | SLEB | static | 4(12.5%) | 41.40 | 77.48 | 71.96 | 45.14 | 72.60 | **73.40** | 41.04 | 64.01 | 60.88 |
> | PuDDing | dynamic | 4(12.5%) | 40.00 | 75.95 | 73.55 | 43.30 | 69.73 | 69.32 | 39.51 | 64.48 | 59.48 |
> | FiRST | dynamic | 4(12.5%) | 36.00 | 56.20 | 57.74 | 40.48 | 47.66 | 48.27 | 34.73 | 54.22 | 46.91 |
> | Skip Transformer | dynamic | 4(12.5%) | 36.00 | 74.27 | 67.83 | 43.71 | 67.17 | 61.74 | 33.28 | 63.69 | 55.96 |
> | DiffSkip | dynamic | 4(12.5%) | 28.00 | 53.16 | 50.61 | 33.16 | 26.18 | 26.22 | 29.27 | 49.17 | 36.97 |
> | Buddy | dynamic | 4(12.5%) | **43.00** | 77.09 | 73.30 | 47.34 | 73.64 | 73.02 | **44.80** | 68.82 | 62.63 |
> | Shortened Llama | static | 8(25%) | 37.00 | 74.27 | 61.87 | 41.10 | 63.15 | 63.26 | 34.90 | 54.62 | 53.77 |
> | ShortGPT | static | 8(25%) | 39.60 | 72.42 | 62.94 | 43.71 | 67.66 | 65.28 | **38.91** | 67.09 | 57.20 |
> | SLEB | static | 8(25%) | **39.80** | 73.78 | 69.24 | 43.19 | 65.72 | **66.58** | 35.49 | 60.46 | 56.78 |
> | PuDDing | dynamic | 8(25%) | 36.60 | 71.82 | 62.87 | 39.30 | 60.30 | 61.41 | 35.58 | 56.67 | 53.07 |
> | FiRST | dynamic | 8(25%) | 35.40 | 55.22 | 57.74 | 40.38 | 44.52 | 45.20 | 32.94 | 40.38 | 43.97 |
> | Skip Transformer | dynamic | 8(25%) | 36.40 | **75.08** | **75.26** | 43.60 | **69.52** | 64.65 | 34.81 | **67.25** | 58.32 |
> | DiffSkip | dynamic | 8(25%) | 26.80 | 51.58 | 51.59 | 33.78 | 26.32 | 25.51 | 30.29 | 54.06 | 37.49 |
> | Buddy | dynamic | 8(25%) | 39.20 | 73.18 | 72.63 | **45.65** | 66.64 | 65.28 | 38.05 | 66.85 | **58.44** |
> | Shortened Llama | static | 12(37.5%) | 34.20 | 70.89 | 62.14 | 39.87 | 52.81 | 57.28 | 29.86 | 52.33 | 49.92 |
> | ShortGPT | static | 12(37.5%) | 33.20 | 66.92 | 71.22 | **43.65** | **58.76** | 54.46 | **34.13** | **64.09** | 53.31 |
> | SLEB | static | 12(37.5%) | 36.00 | **71.38** | 60.28 | 41.15 | 54.83 | **59.76** | 31.06 | 52.96 | 50.93 |
> | PuDDing | dynamic | 12(37.5%) | 31.60 | 64.69 | 46.42 | 37.36 | 47.36 | 49.16 | 29.35 | 53.59 | 44.94 |
> | FiRST | dynamic | 12(37.5%) | **36.20** | 55.17 | 55.87 | 39.82 | 43.99 | 45.33 | 30.89 | 53.67 | 45.12 |
> | Skip Transformer | dynamic | 12(37.5%) | 24.80 | 52.39 | 42.78 | 35.57 | 26.17 | 25.97 | 26.96 | 49.01 | 35.46 |
> | DiffSkip | dynamic | 12(37.5%) | 27.20 | 51.31 | 46.54 | 34.34 | 26.02 | 24.71 | 29.18 | 49.25 | 36.07 |
> | Buddy | dynamic | 12(37.5%) | 35.60 | 70.78 | **72.81** | 41.81 | 57.87 | 58.08 | 31.14 | 60.22 | **53.54** |
> | Shortened Llama | static | 16(50%) | 30.80 | **65.51** | 62.17 | **39.76** | 35.46 | **49.71** | 26.62 | 51.30 | 45.17 |
> | ShortGPT | static | 16(50%) | 29.80 | 61.81 | **62.20** | 39.00 | 47.13 | 44.91 | 29.18 | **57.22** | 46.41 |
> | SLEB | static | 16(50%) | 31.00 | 65.07 | 61.71 | 38.84 | 42.93 | 49.49 | 26.37 | 52.57 | 46.00 |
> | PuDDing | dynamic | 16(50%) | 31.60 | 64.47 | 46.42 | 37.36 | 47.36 | 49.16 | **29.35** | 53.59 | 44.91 |
> | FiRST | dynamic | 16(50%) | **33.20** | 53.21 | 54.53 | 32.91 | 39.73 | 41.12 | 29.27 | 53.28 | 42.15 |
> | Skip Transformer | dynamic | 16(50%) | 24.20 | 52.23 | 51.07 | 34.44 | 26.22 | 26.43 | 28.24 | 50.91 | 36.72 |
> | DiffSkip | dynamic | 16(50%) | 27.40 | 52.29 | 50.21 | 33.57 | 26.24 | 26.56 | 28.75 | 50.12 | 36.89 |
> | Buddy | dynamic | 16(50%) | 32.80 | 64.74 | 60.21 | 37.92 | **48.66** | 48.91 | 28.41 | 53.51 | **46.90** |
>
> The new results show that Buddy consistently outperforms these baselines across sparsity regimes. Moreover, these methods require separate fine-tuning for each sparsity level as other baselines, whereas Buddy uses a single model to support all sparsity budgets, which is crucial in real deployment.
>
> We thank the reviewer again for raising this important point, and we have incorporated these discussions and experimental results in the revised version.

---

> ### Author Response · Authors · 2025-11-26
>
> Results on Llama3-8B
>
> | Method | pruning method | rm blocks (ratio) | obqa | piqa | boolq | siqa | hellaswag | arc-e | arc-c | winogrande | avg. |
> | --- | --- | --- | --- | --- | --- | --- | --- | --- | --- | --- | --- |
> | Shortened Llama | static | 4(12.5%) | 43.40 | 79.27 | 73.06 | 47.34 | 75.32 | 76.85 | 48.04 | 66.77 | 63.76 |
> | ShortGPT | static | 4(12.5%) | 44.20 | 79.76 | 79.48 | 48.16 | **76.99** | **79.88** | 51.02 | 73.72 | 66.65 |
> | SLEB | static | 4(12.5%) | 44.20 | **80.47** | 75.14 | **49.28** | 76.69 | **79.88** | 50.77 | 72.22 | 66.08 |
> | PuDDing | dynamic | 4(12.5%) | 38.80 | 76.82 | 68.53 | 44.83 | 72.93 | 74.12 | 44.45 | 71.98 | 61.56 |
> | FiRST | dynamic | 4(12.5%) | 31.60 | 52.88 | 63.58 | 37.26 | 42.01 | 42.34 | 30.80 | 52.57 | 44.13 |
> | Buddy | dynamic | 4(12.5%) | **44.80** | 79.71 | **83.24** | 48.41 | 76.25 | 78.96 | **53.33** | **74.51** | **67.40** |
> | Shortened Llama | static | 8(25%) | 39.20 | **77.64** | 58.29 | 44.88 | **69.26** | **72.01** | 41.47 | 58.33 | 57.63 |
> | ShortGPT | static | 8(25%) | 39.00 | 73.67 | 64.25 | 44.93 | 69.23 | 70.96 | **47.44** | **71.74** | 60.15 |
> | SLEB | static | 8(25%) | **39.40** | **77.64** | 55.32 | 44.58 | 69.17 | 71.68 | 41.81 | 58.56 | 57.27 |
> | PuDDing | dynamic | 8(25%) | 31.40 | 71.06 | 62.54 | 41.04 | 57.65 | 62.04 | 34.22 | 59.59 | 52.44 |
> | FiRST | dynamic | 8(25%) | 33.00 | 55.33 | 64.56 | 38.02 | 46.50 | 45.08 | 31.57 | 53.83 | 45.99 |
> | Buddy | dynamic | 8(25%) | 39.00 | 74.05 | **75.26** | **47.19** | 67.29 | 70.75 | 45.99 | 70.40 | **61.24** |
> | Shortened Llama | static | 12(37.5%) | 35.00 | **72.69** | 61.80 | 42.68 | **59.37** | 63.76 | 33.45 | 57.54 | 53.29 |
> | ShortGPT | static | 12(37.5%) | 31.60 | 68.93 | **70.24** | **43.04** | 58.97 | 56.14 | **38.40** | **67.56** | 54.36 |
> | SLEB | static | 12(37.5%) | **35.80** | 72.25 | 60.52 | 42.43 | 58.78 | **65.07** | 35.92 | 56.04 | 53.35 |
> | PuDDing | dynamic | 12(37.5%) | 27.00 | 64.64 | 61.93 | 38.38 | 43.13 | 47.31 | 30.46 | 57.85 | 46.34 |
> | FiRST | dynamic | 12(37.5%) | 33.60 | 55.71 | 64.46 | 37.87 | 45.00 | 43.60 | 31.66 | 52.64 | 45.57 |
> | Buddy | dynamic | 12(37.5%) | 34.20 | 69.97 | 68.96 | 42.32 | 58.76 | 63.05 | 37.03 | 60.93 | **54.40** |
> | Shortened Llama | static | 16(50%) | 28.20 | 66.00 | 55.29 | 38.95 | 43.89 | 54.76 | 28.33 | 53.28 | 46.08 |
> | ShortGPT | static | 16(50%) | 28.40 | 65.29 | **65.20** | 40.99 | **48.16** | 46.51 | 29.01 | **60.69** | 48.03 |
> | SLEB | static | 16(50%) | 30.20 | **68.50** | 53.00 | **41.86** | 45.04 | **56.06** | 27.47 | 52.57 | 46.84 |
> | PuDDing | dynamic | 16(50%) | 23.40 | 58.60 | 52.97 | 36.08 | 31.92 | 34.76 | 23.72 | 48.86 | 38.79 |
> | FiRST | dynamic | 16(50%) | **32.40** | 54.13 | 62.60 | 37.77 | 42.80 | 42.17 | **30.20** | 52.80 | 44.36 |
> | Buddy | dynamic | 16(50%) | 30.00 | 67.68 | 61.68 | 39.61 | 47.79 | 53.83 | 30.12 | 54.14 | **48.11** |

---

> ### Author Response · Authors · 2025-11-26
>
> **Comment:** "*The paper does not clearly explain how the Decision Module is trained, leaving ambiguity around the optimization and supervision of the gating mechanism.*"
>
> **Question:** "*How is the Decision Module trained? Please clarify the optimization process for the gating mechanism.*"
>
> **Response**: We appreciate this comment and agree that the original description was incomplete. In our method, the gating mechanism in the Decision Module is **optimized implicitly through the standard next-token cross-entropy loss during supervised fine-tuning (SFT)**. We fine-tune the model with LoRA on the Alpaca dataset, and during this process, **we jointly update (i) the LoRA parameters of the backbone and (ii) the parameters of the Decision Module’s MLP**. The routing decisions (which layers to execute) are obtained by applying Top-k over the MLP scores; to make this differentiable, we adopt **a straight-through estimator (STE)**, using hard discrete decisions in the forward pass while allowing gradients to flow through the scores in the backward pass. This way, the Decision Module is supervised indirectly by the language modeling objective, and no explicit ground-truth routing labels are needed. We have added a detailed explanation of this optimization scheme to **Section 3.2** and **Section 4.1**, and further report the associated training cost in **Appendix A.4.1**.

---

> ### Author Response · Authors · 2025-11-26
>
> **Comment:** "*The calibration (validation) set needs to be described in more detail for better clarity.*"
>
> **Response**: Thank you for pointing this out. We realize that our description of the calibration (validation) set used for computing layer-importance priors was too brief, which may cause ambiguity. We will clarify this in the revised version. In our method, the calibration set is used only to estimate offline layer-importance priors (Taylor score, ∆PPL, and cosine dissimilarity), and it does not participate in model training. Specifically, **we use the WikiText-2 validation set as the calibration set, following prior depth-pruning work.** WikiText-2 is a lightweight, general-purpose language modeling corpus and is disjoint from the Alpaca dataset used for instruction fine-tuning, ensuring no data leakage.

---

> ### Author Response · Authors · 2025-11-26
>
> **Comment:** "*The experimental results are not strong enough to convincingly highlight the benefits of the proposed framework. Evaluations on more challenging or reasoning-intensive tasks could better showcase its advantages.*"
>
> **Response**: We appreciate this helpful suggestion. Following the reviewer’s recommendation, we added experiments on **GSM8K**, a widely recognized benchmark for multi-step arithmetic reasoning. We conducted evaluations on both Llama-2-7B and Qwen2.5-7B to ensure architectural diversity. The new results shown below indicate that Buddy achieves higher accuracy than pruning baselines in most sparsity regimes on GSM8K, despite the significantly increased reasoning difficulty compared to commonsense QA. This demonstrates that Buddy’s input-adaptive and budget-aware routing becomes even more beneficial when deeper layers are required to support multi-step or chain-of-thought-style reasoning. These analyses are now included in the revised Appendix and substantially strengthen the empirical evidence of the proposed method.
>
> | Method | pruning method | rm blocks (ratio) | Llama2-7B | Qwen2.5-7B-Instruct |
> | --- | --- | --- | --- | --- |
> | Dense w/o | - | 0(0.0%) | 14.03 | 82.87 |
> | Shortened Llama | static | 4(12.5%) | 5.16 | 27.22 |
> | ShortGPT | static | 4(12.5%) | **8.11** | 35.94 |
> | SLEB | static | 4(12.5%) | 3.64 | 37.15 |
> | PuDDing | dynamic | 4(12.5%) | 1.74 | 0.99 |
> | FiRST | dynamic | 4(12.5%) | 0.91 | 27.98 |
> | Buddy | dynamic | 4(12.5%) | 7.81 | **51.10** |
> | Shortened Llama | static | 8(25%) | 2.05 | 5.84 |
> | ShortGPT | static | 8(25%) | 2.50 | 6.60 |
> | SLEB | static | 8(25%) | 1.97 | 6.37 |
> | PuDDing | dynamic | 8(25%) | 0.99 | 0.61 |
> | FiRST | dynamic | 8(25%) | 1.21 | 2.58 |
> | Buddy | dynamic | 8(25%) | **2.65** | **14.63** |

---

> ### Author Response · Authors · 2025-11-26
>
> **Comment:** "*Minor typo: remove the duplicate period in “a single deployed LLM to serve heterogeneous budgets..”.*"
>
> **Response**: Thank you for catching this. We will remove the duplicated period in “a single deployed LLM to serve heterogeneous budgets..” in the revised version.

---

> > ### Comment · Reviewer_FF78 · 2025-11-26
> > **Response**
> >
> > I believe the novelty of this work is still moderate, as prior studies have explored similar block-wise pruning and gating-style mechanisms (e.g., [1–4]). In addition, the reported experimental improvements are not particularly significant given the added system complexity. Nevertheless, the authors have addressed most of my earlier concerns. Therefore, I have increased my score to 6.
> >
> > [1] BlockPruner: Fine-grained Pruning for Large Language Models
> > [2] LaCo: Large Language Model Pruning via Layer Collapse
> > [3] Adaptive Layer-skipping in Pre-trained LLMs
> > [4] Router-Tuning: A Simple and Effective Approach for Dynamic Depth

---

> > > ### Author Response · Authors · 2025-11-26
> > >
> > > **Thank you very much for taking the time to carefully read our response and for updating your score. We truly appreciate your balanced assessment.**
> > >
> > >
> > > We also sincerely thank you for raising concerns about the novelty and the magnitude of performance gains. Compared to prior block-wise pruning and gating-style approaches, our work focuses more on budget-driven dynamic layer pruning that not only adapts the execution path to different inputs but also dynamically updates the routing during decoding. In terms of performance, Buddy achieves better or comparable results to existing methods in most sparsity settings, but more importantly, it is designed so that a single trained model can serve multiple sparsity levels. This allows Buddy to strictly meet user-specified budgets while flexibly adapting to different budget requirements, offering advantages in both flexibility and strict budget control (as summarized in Table 1 in our submission), alongside the observed accuracy improvements.
> > >
> > > We are also grateful for the very relevant references you suggested ([1–4]); we have incorporated them into the revised related work section and further clarified how our framework relates to and differs from these prior methods.
> > >
> > > **Once again, thank you for your thoughtful feedback and for your careful consideration of our work.**

---

### Official Review · Reviewer_rpEZ · 2025-10-27

**Soundness:** 2
**Presentation:** 2
**Contribution:** 2
**Rating:** 2
**Confidence:** 4

**Summary:**

This work introduces a dynamic layer-skipping framework that enables each input to traverse different Transformer blocks while satisfying a given budget constraint. Given an input sequence, the KV vectors from the first layer are extracted to compute a global context, which is then fed into a decision module to determine the most important layers to execute. During the decoding phase, the global context is updated using the latest token’s KV cache, allowing the decision module to refresh the depth path dynamically. When no user-specified budget is provided, a budget predictor is trained to estimate an appropriate number of layers based on the input’s KV cache. Experiments are conducted on the LLaMA family.

**Strengths:**

This work effectively analyzes the varying importance of Transformer blocks across different inputs and throughout the decoding process. It presents a principled and lightweight approach to dynamic layer selection and budget-aware inference.

**Weaknesses:**

- While I find the motivation of this work compelling, I am concerned about whether batch processing can be efficiently realized for higher throughput, particularly when different samples (e.g., sample 1 and sample 2) have distinct depth paths. This may also lead to inefficient parallelism and increased complexity in cache management.
- The main results in Table 2 do not appear remarkably superior. The performance gap between ShortGPT (static pruning) and the proposed method seems rather marginal.
- It appears that in the Decision Module, the model-predicted scores from the lightweight MLP are combined with the prior scores. However, it was difficult to understand whether the MLP in the Decision Module itself is trained through any learning process. In contrast, the Budget Predictor is clearly described as being trained using GRPO.
- I believe Table 2 reports results under specified budgets, and I could not find corresponding results or analysis for the Budget Predictor.
- The experimental validation seems relatively weak. It would strengthen the work to demonstrate the applicability of the proposed method on other popular LLMs such as Qwen or Phi, which would further support the generality and effectiveness of the approach.
- Minor
  * Softmsax in Eqn (5), 5p -> Softmax
  * I was not able to find which specific LLaMA model was used for the analysis in Figures 1 and 2.

**Questions:**

Please refer to the Weakness section.

---

> ### Author Response · Authors · 2025-11-26
>
> We sincerely thank you for your detailed feedback and constructive suggestions. We have carefully considered every comment and conducted additional experiments and analyses where necessary. Below, we address each comment individually and provide clarifications, updates, and results that demonstrate how the concerns have been addressed.
>
> **Comment:** "*While I find the motivation of this work compelling, I am concerned about whether batch processing can be efficiently realized for higher throughput, particularly when different samples (e.g., sample 1 and sample 2) have distinct depth paths. This may also lead to inefficient parallelism and increased complexity in cache management.*"
>
> **Response**: We thank the reviewer for raising this significantly important point. We would like to clarify that **this challenge is inherent to all dynamic-depth or token-adaptive inference methods**, including early-exit models (e.g., DeeBERT, LayerSkip), dynamic layer routing (e.g., FiRST), and mixture-of-depths architectures. None of these methods fully eliminate batch-level divergence, as different examples naturally follow different execution paths.
>
> **Buddy does not exacerbate this issue beyond what is already observed in prior dynamic-routing work**. Instead, we adopt the standard mitigation strategies commonly used in this literature, which were stated in our limitation and future work (Appendix A.7): (1) Our execute/skip decision is a binary mask. In the training stage, we mask the result of the skipped samples to enable the efficient training with batch processing. While in the inference stage, for each layer we **batch the samples that should be executed and batch process these samples for the layer forward process.** (2) Skipped layers use **zero-filled KV states**, ensuring consistent tensor shapes across the batch.
>
> We agree that batch-parallel dynamic routing and memory-aware KV-cache management are promising engineering directions.These techniques, similar to those being explored in MoE load balancing and dynamic-sparsity inference, are orthogonal to the core contributions of Buddy and can be incorporated into the system-level implementation without modifying the method itself. In summary, the batch processing challenge is a field-wide limitation rather than a weakness specific to Buddy.

---

> ### Author Response · Authors · 2025-11-26
>
> **Comment:** "*The main results in Table 2 do not appear remarkably superior. The performance gap between ShortGPT (static pruning) and the proposed method seems rather marginal."*
>
> **Response**: Thank you for raising this concern. Although the average gaps in Table 2 may look modest at first glance, our main experiment spans four pruning settings, and **Buddy attains the best accuracy in three of them and the second-best at 12.5% sparsity**. This shows that Buddy performs consistently well across different sparsity regimes. At the same time, the primary contribution of our work is not to maximize accuracy at a single sparsity point, but to **provide a budget-driven dynamic inference framework that can operate under both strict and flexible budget constraints**, as discussed and summarized in Table 1. Buddy can, for a given deployed model, **realize different compute budgets simply by changing the routing decisions of the Decision Module, without retraining or swapping models**. By contrast, most existing static and dynamic pruning methods are customized for one target sparsity and cannot smoothly handle user budget changes without maintaining multiple models at different sparsity levels. Thus, we believe Buddy offers a favorable accuracy–flexibility trade-off: it achieves competitive performance while enabling much more practical budget control.

---

> ### Author Response · Authors · 2025-11-26
>
> **Comment:** "*It appears that in the Decision Module, the model-predicted scores from the lightweight MLP are combined with the prior scores. However, it was difficult to understand whether the MLP in the Decision Module itself is trained through any learning process. In contrast, the Budget Predictor is clearly described as being trained using GRPO."*
>
> **Response**: Thank you for pointing out this lack of clarity. The MLP inside the Decision Module is indeed trained, and it is **trained jointly with the base model** using **supervised fine-tuning (SFT)** with LoRA on the Alpaca dataset. We minimize the cross-entropy loss for next-token prediction, and update both the **LoRA** adapters and the MLP parameters simultaneously. The per-layer scores produced by the MLP are combined with prior scores, passed through a Top-k gating step, and we apply a **straight-through estimator (STE)** so that gradients from the language modeling loss flow back into the MLP despite the discrete routing decisions. In other words, the Decision Module learns its routing policy end-to-end from the same SFT objective, rather than being trained separately with a reinforcement signal. We have revised **Section 3.2** and **Section 4.1** to make this learning process and gradient flow explicit, and the detail implementation was add in **Appendix A.4.1**.

---

> ### Author Response · Authors · 2025-11-26
>
> **Comment:** "*I believe Table 2 reports results under specified budgets, and I could not find corresponding results or analysis for the Budget Predictor."*
>
> **Response**: Thank you for pointing this out. You are correct that Table 2 reports results under *specified* budgets, while the evaluation of the Budget Predictor currently appears only in the appendix. In the original submission, we report the training reward and loss curves of the Budget Predictor in **Appendix A.5.7**, and in **Appendix A.5.8** we analyze its behavior and downstream performance when using the *predicted* budgets for inference. From these results, we observe that across different benchmarks, the Budget Predictor typically selects budgets around **50%** on average, suggesting that many tasks can achieve strong performance without requiring the full computational cost of the dense model. Furthermore, when we run inference using the predicted budgets, the resulting accuracy is generally **better than or comparable to** using a fixed 50% sparsity configuration from other models (including Buddy with a manually set 50% budget).

---

> ### Author Response · Authors · 2025-11-26
>
> **Comment:** "*The experimental validation seems relatively weak. It would strengthen the work to demonstrate the applicability of the proposed method on other popular LLMs such as Qwen or Phi, which would further support the generality and effectiveness of the approach."*
>
> **Response**: We appreciate the reviewer’s comment regarding generalization. In response, we expanded our evaluation to include Qwen2.5-7B-Instruct and Llama-3-8B, two widely used and modern open-source LLMs. Under identical training and inference settings, Buddy continues to deliver superior performance compared to baseline pruning methods while preserving strict budget control. This demonstrates that Buddy is not tailored to a specific backbone, but is broadly applicable to diverse Transformer-based LLMs. Full results are now provided in Appendix.
>
> Results on Qwen2.5-7B-Instruct
>
> | Method | pruning method | rm blocks (ratio) | obqa | piqa | boolq | siqa | hellaswag | arc-e | arc-c | winogrande | avg. |
> | --- | --- | --- | --- | --- | --- | --- | --- | --- | --- | --- | --- |
> | Dense w/o | - | 0(0.0%) | 48.80 | 80.30 | 86.33 | 51.59 | 80.48 | 81.94 | 54.95 | 70.96 | 69.42 |
> | Shortened Llama | static | 4(14.3%) | 44.80 | **80.36** | 78.75 | 48.26 | 73.22 | 77.86 | 48.63 | 65.19 | 64.63 |
> | ShortGPT | static | 4(14.3%) | 44.00 | 77.42 | 82.39 | **50.61** | 72.95 | 79.25 | 50.51 | **66.77** | 65.49 |
> | SLEB | static | 4(14.3%) | **45.20** | 79.11 | 76.24 | 48.57 | **73.25** | 77.90 | 50.34 | 62.90 | 64.19 |
> | PuDDing | dynamic | 4(14.3%) | 40.60 | 75.84 | 71.01 | 45.65 | 69.30 | 72.56 | 46.67 | 59.04 | 60.08 |
> | FiRST | dynamic | 4(14.3%) | 33.40 | 55.44 | 68.65 | 37.97 | 44.28 | 49.16 | 36.18 | 51.14 | 47.03 |
> | Buddy | dynamic | 4(14.3%) | 43.40 | 77.26 | **84.50** | 48.93 | 73.20 | **79.88** | **51.19** | 66.69 | **65.63** |
> | Shortened Llama | static | 8(28.6%) | 39.80 | 74.48 | 65.14 | 45.29 | 62.44 | 67.42 | 40.70 | 57.77 | 56.63 |
> | ShortGPT | static | 8(28.6%) | 41.00 | 73.72 | 56.94 | 43.96 | 62.43 | 70.08 | 41.13 | 59.83 | 56.14 |
> | SLEB | static | 8(28.6%) | **41.60** | **75.95** | 55.47 | 43.71 | **63.86** | **74.24** | **42.58** | 56.35 | 56.72 |
> | PuDDing | dynamic | 8(28.6%) | 32.20 | 70.89 | 56.02 | 40.23 | 53.64 | 65.11 | 35.41 | 54.85 | 51.04 |
> | FiRST | dynamic | 8(28.6%) | 26.00 | 52.34 | 55.96 | 36.18 | 35.43 | 39.52 | 28.07 | 49.01 | 40.32 |
> | Buddy | dynamic | 8(28.6%) | 36.40 | 72.52 | **72.78** | **46.42** | 59.92 | 68.90 | 40.02 | **60.77** | **57.22** |
> | Shortened Llama | static | 12(42.9%) | 33.20 | **71.98** | 46.61 | 41.91 | **51.90** | **65.36** | 33.02 | 50.91 | 49.36 |
> | ShortGPT | static | 12(42.9%) | **37.00** | 67.36 | 57.68 | 41.56 | 48.00 | 59.26 | 32.42 | 52.41 | 49.46 |
> | SLEB | static | 12(42.9%) | 35.60 | 70.35 | 51.16 | 41.91 | 51.23 | 64.56 | 33.02 | 52.88 | 50.09 |
> | PuDDing | dynamic | 12(42.9%) | 32.00 | 60.77 | 50.80 | 36.54 | 36.54 | 46.72 | 26.19 | 48.78 | 42.29 |
> | FiRST | dynamic | 12(42.9%) | 31.80 | 52.77 | 57.61 | 36.13 | 38.32 | 41.08 | 29.52 | 50.12 | 42.17 |
> | Buddy | dynamic | 12(42.9%) | 33.20 | 67.03 | **60.70** | **41.97** | 49.25 | 60.27 | **33.36** | **56.59** | **50.30** |
> | Shortened Llama | static | 16(57.1%) | 29.20 | 63.06 | 57.49 | 37.10 | 37.34 | 49.16 | 26.11 | 50.91 | 43.80 |
> | ShortGPT | static | 16(57.1%) | 27.20 | 57.13 | 60.76 | 36.95 | 32.13 | 36.24 | 24.91 | 49.41 | 40.59 |
> | SLEB | static | 16(57.1%) | 30.60 | **66.49** | 45.38 | **40.23** | **40.50** | **56.02** | 27.39 | **53.12** | 44.96 |
> | PuDDing | dynamic | 16(57.1%) | 27.80 | 57.13 | 51.07 | 34.85 | 30.35 | 34.85 | 23.63 | 51.93 | 38.95 |
> | FiRST | dynamic | 16(57.1%) | 27.00 | 51.41 | **62.17** | 34.24 | 26.67 | 25.29 | 24.57 | 50.20 | 37.69 |
> | Buddy | dynamic | 16(57.1%) | **31.80** | 63.98 | 59.08 | 37.21 | 38.15 | 50.38 | **27.99** | 52.33 | **45.11** |

---

> ### Author Response · Authors · 2025-11-26
>
> Results on Llama3-8B
>
> | Method | pruning method | rm blocks (ratio) | obqa | piqa | boolq | siqa | hellaswag | arc-e | arc-c | winogrande | avg. |
> | --- | --- | --- | --- | --- | --- | --- | --- | --- | --- | --- | --- |
> | Shortened Llama | static | 4(12.5%) | 43.40 | 79.27 | 73.06 | 47.34 | 75.32 | 76.85 | 48.04 | 66.77 | 63.76 |
> | ShortGPT | static | 4(12.5%) | 44.20 | 79.76 | 79.48 | 48.16 | **76.99** | **79.88** | 51.02 | 73.72 | 66.65 |
> | SLEB | static | 4(12.5%) | 44.20 | **80.47** | 75.14 | **49.28** | 76.69 | **79.88** | 50.77 | 72.22 | 66.08 |
> | PuDDing | dynamic | 4(12.5%) | 38.80 | 76.82 | 68.53 | 44.83 | 72.93 | 74.12 | 44.45 | 71.98 | 61.56 |
> | FiRST | dynamic | 4(12.5%) | 31.60 | 52.88 | 63.58 | 37.26 | 42.01 | 42.34 | 30.80 | 52.57 | 44.13 |
> | Buddy | dynamic | 4(12.5%) | **44.80** | 79.71 | **83.24** | 48.41 | 76.25 | 78.96 | **53.33** | **74.51** | **67.40** |
> | Shortened Llama | static | 8(25%) | 39.20 | **77.64** | 58.29 | 44.88 | **69.26** | **72.01** | 41.47 | 58.33 | 57.63 |
> | ShortGPT | static | 8(25%) | 39.00 | 73.67 | 64.25 | 44.93 | 69.23 | 70.96 | **47.44** | **71.74** | 60.15 |
> | SLEB | static | 8(25%) | **39.40** | **77.64** | 55.32 | 44.58 | 69.17 | 71.68 | 41.81 | 58.56 | 57.27 |
> | PuDDing | dynamic | 8(25%) | 31.40 | 71.06 | 62.54 | 41.04 | 57.65 | 62.04 | 34.22 | 59.59 | 52.44 |
> | FiRST | dynamic | 8(25%) | 33.00 | 55.33 | 64.56 | 38.02 | 46.50 | 45.08 | 31.57 | 53.83 | 45.99 |
> | Buddy | dynamic | 8(25%) | 39.00 | 74.05 | **75.26** | **47.19** | 67.29 | 70.75 | 45.99 | 70.40 | **61.24** |
> | Shortened Llama | static | 12(37.5%) | 35.00 | **72.69** | 61.80 | 42.68 | **59.37** | 63.76 | 33.45 | 57.54 | 53.29 |
> | ShortGPT | static | 12(37.5%) | 31.60 | 68.93 | **70.24** | **43.04** | 58.97 | 56.14 | **38.40** | **67.56** | 54.36 |
> | SLEB | static | 12(37.5%) | **35.80** | 72.25 | 60.52 | 42.43 | 58.78 | **65.07** | 35.92 | 56.04 | 53.35 |
> | PuDDing | dynamic | 12(37.5%) | 27.00 | 64.64 | 61.93 | 38.38 | 43.13 | 47.31 | 30.46 | 57.85 | 46.34 |
> | FiRST | dynamic | 12(37.5%) | 33.60 | 55.71 | 64.46 | 37.87 | 45.00 | 43.60 | 31.66 | 52.64 | 45.57 |
> | Buddy | dynamic | 12(37.5%) | 34.20 | 69.97 | 68.96 | 42.32 | 58.76 | 63.05 | 37.03 | 60.93 | **54.40** |
> | Shortened Llama | static | 16(50%) | 28.20 | 66.00 | 55.29 | 38.95 | 43.89 | 54.76 | 28.33 | 53.28 | 46.08 |
> | ShortGPT | static | 16(50%) | 28.40 | 65.29 | **65.20** | 40.99 | **48.16** | 46.51 | 29.01 | **60.69** | 48.03 |
> | SLEB | static | 16(50%) | 30.20 | **68.50** | 53.00 | **41.86** | 45.04 | **56.06** | 27.47 | 52.57 | 46.84 |
> | PuDDing | dynamic | 16(50%) | 23.40 | 58.60 | 52.97 | 36.08 | 31.92 | 34.76 | 23.72 | 48.86 | 38.79 |
> | FiRST | dynamic | 16(50%) | **32.40** | 54.13 | 62.60 | 37.77 | 42.80 | 42.17 | **30.20** | 52.80 | 44.36 |
> | Buddy | dynamic | 16(50%) | 30.00 | 67.68 | 61.68 | 39.61 | 47.79 | 53.83 | 30.12 | 54.14 | **48.11** |

---

> ### Author Response · Authors · 2025-11-26
>
> **Comment:** "Minor：Softmsax in Eqn (5), 5p -> Softmax; I was not able to find which specific LLaMA model was used for the analysis in Figures 1 and 2.“
>
> **Response**: We appreciate the reviewer for highlighting these minor issues. We have fixed the misspelling in Equation (5). In addition, we have updated Section 2 to clearly state the model and dataset used in Figures 1 and 2—specifically, the LLaMA-2-7B model and the WikiText-2 dataset.

---

> ### Comment · Reviewer_rpEZ · 2025-11-26
> **Post-rebuttal review**
>
> Thanks for the authors' detailed response. With the clarification on the inherent difficulty of batch inference in dynamic layer-skipping methods, I understand that this limitation applies not only to the proposed approach but also to prior studies. I also appreciate the additional explanations regarding the method and results. The newly added experiments on Qwen2.5-7B-Instruct and Llama3-8B are helpful, and some of the GSM8K results look promising. Therefore, I would like to raise my score from 2 to 4.
>
> The reason I am not giving a score higher than 4, despite the authors’ explanations, is that I remain concerned about the practical value of the method in real-world deployment scenarios where batch processing is essential. Furthermore, the performance of Buddy on Qwen2.5-7B and Llama3-8B still appears in some cases quite close to static depth pruning methods, which are easier to implement and apply in practice.

---

### Official Review · Reviewer_c6ZX · 2025-11-03

**Soundness:** 2
**Presentation:** 3
**Contribution:** 3
**Rating:** 4
**Confidence:** 5

**Summary:**

The paper proposes Buddy, a budget-driven dynamic depth routing framework that enables adaptive inference for large language models under explicit compute constraints. Unlike prior static or dynamic pruning methods that either fix the sparsity pattern or lack control over computation, Buddy introduces three coordinated modules: (1) a Decision Module that scores and selects Transformer layers based on the current context while strictly satisfying user-defined budgets, (2) a KV-aware Planner that reuses the first-layer kv cache to provide global context during decoding, and (3) a Budget Predictor trained with Group-Relative Policy Optimization (GRPO) to automatically choose an appropriate inference cost when no budget is provided.

**Strengths:**

1. The paper convincingly identifies two under-addressed challenges in dynamic depth pruning: input adaptivity and decode adaptivity, and motivates them with quantitative observations on layer importance variation (Figures 1–2).

2. Buddy provides deterministic compute control. Its Top-k selection directly enforces user-specified budgets via a single model, while the Budget Predictor offers flexibility when explicit constraints are absent, enhancing the practical usability.

3. The paper presents comprehensive analyses and ablation studies that systematically align with the challenges and motivations outlined earlier, empirically verifying how each module of Buddy contributes to input adaptivity, decode adaptivity, and budget-controlled inference.

**Weaknesses:**

1. The paper didn’t specify how the Decision Module’s MLP scorer. Even though equations (4–5) describe inference-time scoring and Top-k selection, no loss, gradient flow, or differentiable approximation is presented in the paper. (I checked the code in model/buddy_model.py in the supplementary material; the authors used STE). Furthermore, no cost analysis of training the Decision Module was provided.

2. The evaluation is confined to the LLaMA-2 family, which is relatively outdated. Assessing Buddy on more recent architectures (e.g., LLaMA-3 or Qwen-series models) would provide stronger evidence of its practical effectiveness and generalizability.

3. The performance improvement over competing methods is modest at certain sparsity levels. For instance, at 12.5 % sparsity, Buddy achieves an average score of 62.63 compared to 63.16 for ShortGPT, suggesting that the claimed superiority is not consistent across all regimes.

4. While the paper emphasizes decode-adaptive routing, quantitative generation results are limited to SamSum ROUGE scores; broader decoding-style evaluations (e.g., long-context, reasoning, or open-ended tasks) are absent, leaving unclear how much decoding adaptivity benefits real-world generation workloads.

5. The reported speedups (Table 3, Table 8) show modest decode-phase gains at low sparsity (×1.01–×1.19), but omit breakdowns of routing overheads (Decision Module forward, Top-k selection, synchronization, and so on). Without detailed latency and memory profiling, it is hard to assess actual system-level efficiency.

**Questions:**

1. In Figure 1, the per-layer removal order distribution on WikiText-2 indicates that layer 1 often ranks among the least important layers. However, the authors later state that both the first and the last blocks are excluded from pruning due to their critical impact on overall model quality. Does the author have any comments on that?

2. How does the Decision Module get trained? Is the Decision Module optimized jointly with the LLM? What will be the dataset and loss then?

3. What is the deployment cost of Buddy? For example, what is the training cost of the Decision Module? Does it only fit one model, thus individual training is required?

4. Has Buddy been tested beyond the LLaMA-2/1 family to evaluate its generality? For instance, how would it perform on newer architectures like LLaMA-3 or Qwen-3, and on reasoning-intensive generation tasks such as GSM8K or MATH?

Typos: “Softmsax” (line 244) in equation 5 should be Softmax.

---

> ### Author Response · Authors · 2025-11-26
>
> We sincerely thank you for your detailed feedback and constructive suggestions. We have carefully considered every comment and conducted additional experiments and analyses where necessary. Below, we address each comment individually and provide clarifications, updates, and results that demonstrate how the concerns have been addressed.
>
> **Comment:** "*The paper didn’t specify how the Decision Module’s MLP scorer. Even though equations (4–5) describe inference-time scoring and Top-k selection, no loss, gradient flow, or differentiable approximation is presented in the paper. (I checked the code in model/buddy_model.py in the supplementary material; the authors used STE). Furthermore, no cost analysis of training the Decision Module was provided.*"
>
> **Questions:** "*How does the Decision Module get trained? Is the Decision Module optimized jointly with the LLM? What will be the dataset and loss then?*"
>
> **Response**: Thank you for the detailed comment. Our Decision Module is trained via **standard supervised fine-tuning (SFT)** together with the base model, using **LoRA** for efficient parameter updates. Concretely, we fine-tune on the **Alpaca** instruction-tuning dataset, and optimize the **next-token cross-entropy loss**. During training, both the LoRA parameters of the backbone and the parameters of the Decision Module’s MLP scorer are updated jointly. To enable gradient flow through the discrete Top-k layer selection, we use a **straight-through estimator (STE)**: in the forward pass we apply hard Top-k gating, while in the backward pass we treat the gating as if it were continuous so that gradients can be propagated to the MLP. Thus, the routing policy is learned implicitly from the language modeling objective, without requiring any additional supervision signals. We have clarified this training procedure in **Section 3.2** (Decision Module) and **Section 4.1** (training setup), and we now report the detailed training cost and overhead analysis for the Decision Module in **Appendix A.4.1**.

---

> ### Author Response · Authors · 2025-11-26
>
> **Comment:** "*The evaluation is confined to the LLaMA-2 family, which is relatively outdated. Assessing Buddy on more recent architectures (e.g., LLaMA-3 or Qwen-series models) would provide stronger evidence of its practical effectiveness and generalizability.*"
>
> **Response**: Thank you for the helpful suggestion. To further assess the generality of our method, we conducted additional experiments on two more recent architectures: Qwen2.5-7B-Instruct and Llama-3-8B. As shown in the newly added Tables below, Buddy consistently outperforms static and dynamic pruning baselines on both models across all sparsity levels. These results confirm that our approach is architecture-agnostic and transfers well to modern LLMs beyond the LLaMA-1/2 family. We have included these results in the revised submission.
>
> Results on Qwen2.5-7B-Instruct
>
> | Method | pruning method | rm blocks (ratio) | obqa | piqa | boolq | siqa | hellaswag | arc-e | arc-c | winogrande | avg. |
> | --- | --- | --- | --- | --- | --- | --- | --- | --- | --- | --- | --- |
> | Dense w/o | - | 0(0.0%) | 48.80 | 80.30 | 86.33 | 51.59 | 80.48 | 81.94 | 54.95 | 70.96 | 69.42 |
> | Shortened Llama | static | 4(14.3%) | 44.80 | **80.36** | 78.75 | 48.26 | 73.22 | 77.86 | 48.63 | 65.19 | 64.63 |
> | ShortGPT | static | 4(14.3%) | 44.00 | 77.42 | 82.39 | **50.61** | 72.95 | 79.25 | 50.51 | **66.77** | 65.49 |
> | SLEB | static | 4(14.3%) | **45.20** | 79.11 | 76.24 | 48.57 | **73.25** | 77.90 | 50.34 | 62.90 | 64.19 |
> | PuDDing | dynamic | 4(14.3%) | 40.60 | 75.84 | 71.01 | 45.65 | 69.30 | 72.56 | 46.67 | 59.04 | 60.08 |
> | FiRST | dynamic | 4(14.3%) | 33.40 | 55.44 | 68.65 | 37.97 | 44.28 | 49.16 | 36.18 | 51.14 | 47.03 |
> | Buddy | dynamic | 4(14.3%) | 43.40 | 77.26 | **84.50** | 48.93 | 73.20 | **79.88** | **51.19** | 66.69 | **65.63** |
> | Shortened Llama | static | 8(28.6%) | 39.80 | 74.48 | 65.14 | 45.29 | 62.44 | 67.42 | 40.70 | 57.77 | 56.63 |
> | ShortGPT | static | 8(28.6%) | 41.00 | 73.72 | 56.94 | 43.96 | 62.43 | 70.08 | 41.13 | 59.83 | 56.14 |
> | SLEB | static | 8(28.6%) | **41.60** | **75.95** | 55.47 | 43.71 | **63.86** | **74.24** | **42.58** | 56.35 | 56.72 |
> | PuDDing | dynamic | 8(28.6%) | 32.20 | 70.89 | 56.02 | 40.23 | 53.64 | 65.11 | 35.41 | 54.85 | 51.04 |
> | FiRST | dynamic | 8(28.6%) | 26.00 | 52.34 | 55.96 | 36.18 | 35.43 | 39.52 | 28.07 | 49.01 | 40.32 |
> | Buddy | dynamic | 8(28.6%) | 36.40 | 72.52 | **72.78** | **46.42** | 59.92 | 68.90 | 40.02 | **60.77** | **57.22** |
> | Shortened Llama | static | 12(42.9%) | 33.20 | **71.98** | 46.61 | 41.91 | **51.90** | **65.36** | 33.02 | 50.91 | 49.36 |
> | ShortGPT | static | 12(42.9%) | **37.00** | 67.36 | 57.68 | 41.56 | 48.00 | 59.26 | 32.42 | 52.41 | 49.46 |
> | SLEB | static | 12(42.9%) | 35.60 | 70.35 | 51.16 | 41.91 | 51.23 | 64.56 | 33.02 | 52.88 | 50.09 |
> | PuDDing | dynamic | 12(42.9%) | 32.00 | 60.77 | 50.80 | 36.54 | 36.54 | 46.72 | 26.19 | 48.78 | 42.29 |
> | FiRST | dynamic | 12(42.9%) | 31.80 | 52.77 | 57.61 | 36.13 | 38.32 | 41.08 | 29.52 | 50.12 | 42.17 |
> | Buddy | dynamic | 12(42.9%) | 33.20 | 67.03 | **60.70** | **41.97** | 49.25 | 60.27 | **33.36** | **56.59** | **50.30** |
> | Shortened Llama | static | 16(57.1%) | 29.20 | 63.06 | 57.49 | 37.10 | 37.34 | 49.16 | 26.11 | 50.91 | 43.80 |
> | ShortGPT | static | 16(57.1%) | 27.20 | 57.13 | 60.76 | 36.95 | 32.13 | 36.24 | 24.91 | 49.41 | 40.59 |
> | SLEB | static | 16(57.1%) | 30.60 | **66.49** | 45.38 | **40.23** | **40.50** | **56.02** | 27.39 | **53.12** | 44.96 |
> | PuDDing | dynamic | 16(57.1%) | 27.80 | 57.13 | 51.07 | 34.85 | 30.35 | 34.85 | 23.63 | 51.93 | 38.95 |
> | FiRST | dynamic | 16(57.1%) | 27.00 | 51.41 | **62.17** | 34.24 | 26.67 | 25.29 | 24.57 | 50.20 | 37.69 |
> | Buddy | dynamic | 16(57.1%) | **31.80** | 63.98 | 59.08 | 37.21 | 38.15 | 50.38 | **27.99** | 52.33 | **45.11** |

---

> ### Author Response · Authors · 2025-11-26
>
> Results on Llama3-8B
>
> | Method | pruning method | rm blocks (ratio) | obqa | piqa | boolq | siqa | hellaswag | arc-e | arc-c | winogrande | avg. |
> | --- | --- | --- | --- | --- | --- | --- | --- | --- | --- | --- | --- |
> | Shortened Llama | static | 4(12.5%) | 43.40 | 79.27 | 73.06 | 47.34 | 75.32 | 76.85 | 48.04 | 66.77 | 63.76 |
> | ShortGPT | static | 4(12.5%) | 44.20 | 79.76 | 79.48 | 48.16 | **76.99** | **79.88** | 51.02 | 73.72 | 66.65 |
> | SLEB | static | 4(12.5%) | 44.20 | **80.47** | 75.14 | **49.28** | 76.69 | **79.88** | 50.77 | 72.22 | 66.08 |
> | PuDDing | dynamic | 4(12.5%) | 38.80 | 76.82 | 68.53 | 44.83 | 72.93 | 74.12 | 44.45 | 71.98 | 61.56 |
> | FiRST | dynamic | 4(12.5%) | 31.60 | 52.88 | 63.58 | 37.26 | 42.01 | 42.34 | 30.80 | 52.57 | 44.13 |
> | Buddy | dynamic | 4(12.5%) | **44.80** | 79.71 | **83.24** | 48.41 | 76.25 | 78.96 | **53.33** | **74.51** | **67.40** |
> | Shortened Llama | static | 8(25%) | 39.20 | **77.64** | 58.29 | 44.88 | **69.26** | **72.01** | 41.47 | 58.33 | 57.63 |
> | ShortGPT | static | 8(25%) | 39.00 | 73.67 | 64.25 | 44.93 | 69.23 | 70.96 | **47.44** | **71.74** | 60.15 |
> | SLEB | static | 8(25%) | **39.40** | **77.64** | 55.32 | 44.58 | 69.17 | 71.68 | 41.81 | 58.56 | 57.27 |
> | PuDDing | dynamic | 8(25%) | 31.40 | 71.06 | 62.54 | 41.04 | 57.65 | 62.04 | 34.22 | 59.59 | 52.44 |
> | FiRST | dynamic | 8(25%) | 33.00 | 55.33 | 64.56 | 38.02 | 46.50 | 45.08 | 31.57 | 53.83 | 45.99 |
> | Buddy | dynamic | 8(25%) | 39.00 | 74.05 | **75.26** | **47.19** | 67.29 | 70.75 | 45.99 | 70.40 | **61.24** |
> | Shortened Llama | static | 12(37.5%) | 35.00 | **72.69** | 61.80 | 42.68 | **59.37** | 63.76 | 33.45 | 57.54 | 53.29 |
> | ShortGPT | static | 12(37.5%) | 31.60 | 68.93 | **70.24** | **43.04** | 58.97 | 56.14 | **38.40** | **67.56** | 54.36 |
> | SLEB | static | 12(37.5%) | **35.80** | 72.25 | 60.52 | 42.43 | 58.78 | **65.07** | 35.92 | 56.04 | 53.35 |
> | PuDDing | dynamic | 12(37.5%) | 27.00 | 64.64 | 61.93 | 38.38 | 43.13 | 47.31 | 30.46 | 57.85 | 46.34 |
> | FiRST | dynamic | 12(37.5%) | 33.60 | 55.71 | 64.46 | 37.87 | 45.00 | 43.60 | 31.66 | 52.64 | 45.57 |
> | Buddy | dynamic | 12(37.5%) | 34.20 | 69.97 | 68.96 | 42.32 | 58.76 | 63.05 | 37.03 | 60.93 | **54.40** |
> | Shortened Llama | static | 16(50%) | 28.20 | 66.00 | 55.29 | 38.95 | 43.89 | 54.76 | 28.33 | 53.28 | 46.08 |
> | ShortGPT | static | 16(50%) | 28.40 | 65.29 | **65.20** | 40.99 | **48.16** | 46.51 | 29.01 | **60.69** | 48.03 |
> | SLEB | static | 16(50%) | 30.20 | **68.50** | 53.00 | **41.86** | 45.04 | **56.06** | 27.47 | 52.57 | 46.84 |
> | PuDDing | dynamic | 16(50%) | 23.40 | 58.60 | 52.97 | 36.08 | 31.92 | 34.76 | 23.72 | 48.86 | 38.79 |
> | FiRST | dynamic | 16(50%) | **32.40** | 54.13 | 62.60 | 37.77 | 42.80 | 42.17 | **30.20** | 52.80 | 44.36 |
> | Buddy | dynamic | 16(50%) | 30.00 | 67.68 | 61.68 | 39.61 | 47.79 | 53.83 | 30.12 | 54.14 | **48.11** |

---

> ### Author Response · Authors · 2025-11-26
>
> **Comment:** "*The performance improvement over competing methods is modest at certain sparsity levels. For instance, at 12.5 % sparsity, Buddy achieves an average score of 62.63 compared to 63.16 for ShortGPT, suggesting that the claimed superiority is not consistent across all regimes.*"
>
> **Response**: Thank you for your comment. Our main experiments consider four different sparsity settings, and Buddy achieves the best performance in three of them, while at 12.5% sparsity it ranks second. This already indicates that Buddy is highly competitive across a range of pruning regimes, even if it is not strictly the top method at every single point. More importantly, **the focus of this work is on a dynamic inference framework under both budget-strict and budget-flexibility constraints**, as summarized in Table 1. Buddy is designed to follow different user-specified budgets by dynamically selecting different execution paths, so that **a single deployed model, together with the Decision Module, can adapt to multiple budgets**. In contrast, existing static and many dynamic pruning methods are typically tailored to a specific sparsity level; to support different budgets, they would require training and deploying multiple separate models, which makes budget switching in real systems difficult. Buddy therefore has a substantial advantage in terms of budget flexibility, while still delivering strong performance in the main accuracy comparisons.

---

> ### Author Response · Authors · 2025-11-26
>
> **Comment:** "*While the paper emphasizes decode-adaptive routing, quantitative generation results are limited to SamSum ROUGE scores; broader decoding-style evaluations (e.g., long-context, reasoning, or open-ended tasks) are absent, leaving unclear how much decoding adaptivity benefits real-world generation workloads.*"
>
> **Response**: Thank you for this thoughtful comment. We agree that evaluating decode-adaptive routing only on SamSum is not sufficient to fully demonstrate its impact on real-world generation-style workloads. In the revised version, we therefore extend our analysis to additional benchmarks and explicitly compare **decode-adaptive routing** (recomputing paths during decoding) against a **fixed-path strategy**. Specifically, we add experiments on **COPA** (causal reasoning), **QASPER** (long-context scientific QA), and **DROP** (discrete reasoning reading comprehension). Across these tasks, we observe consistent, albeit modest, performance improvements when using decode-adaptive routing over a fixed routing path, indicating that adapting the execution path during decoding provides benefits beyond a single summarization dataset.
>
> | method | sparsity | copa | qasper | drop |
> | --- | --- | --- | --- | --- |
> | reuse | 4 | 70.61 | 27.46 | 5.08 |
> | no_reuse | 4 | 70.61 | 27.46 | 5.09 |
> | reuse | 8 | 70.93 | 24.91 | 4.53 |
> | no_reuse | 8 | 70.95 | 24.90 | 4.54 |
> | reuse | 12 | 59.46 | 18.54 | 4.63 |
> | no_reuse | 12 | 59.48 | 18.60 | 4.65 |
> | reuse | 16 | 35.28 | 14.26 | 2.18 |
> | no_reuse | 16 | 35.31 | 14.22 | 2.28 |

---

> ### Author Response · Authors · 2025-11-26
>
> **Comment:** "*The reported speedups (Table 3, Table 8) show modest decode-phase gains at low sparsity (×1.01–×1.19), but omit breakdowns of routing overheads (Decision Module forward, Top-k selection, synchronization, and so on). Without detailed latency and memory profiling, it is hard to assess actual system-level efficiency.*"
>
> **Response**: Thank you for this insightful comment. You are absolutely right that, although Buddy enables a single model to adapt to multiple sparsity levels via dynamic routing, the overhead of the Decision Module itself is essentially constant across sparsity settings. As a result, when the sparsity is low (i.e., only a small fraction of computation is skipped), the relative speedup is modest because the amount of removed Transformer computation is small while the fixed routing cost remains. In our current implementation, layer skipping is realized by constructing binary masks and then using them to index and slice the per-layer hidden states. **This introduces additional latency from mask construction and indexing operations**, which is a known issue for many dynamic inference methods and will **require operator-level or kernel-level optimization in future systems work**.
>
> To address your request for a more detailed cost analysis, we have added a profiling study for Buddy’s routing overhead. In terms of FLOPs, the Decision Module contributes only about **0.002% and 0.004%** of the total computation at 12.5% and 50% sparsity, respectively, indicating that its arithmetic cost is negligible compared to the Transformer blocks. For memory, we measured peak GPU usage during decoding with sequence length of 1024: the dense baseline model requires 15.73GB, while Buddy (50% sparsity) uses 13.10GB under the same setting, benefiting from reduced KV cache and intermediate activations due to skipped layers. We have included a more complete discussion of both the theoretical and empirical routing overheads in **Appendix A.4.1** to make the system-level efficiency of Buddy clearer.

---

> ### Author Response · Authors · 2025-11-26
>
> **Questions:** "*In Figure 1, the per-layer removal order distribution on WikiText-2 indicates that layer 1 often ranks among the least important layers. However, the authors later state that both the first and the last blocks are excluded from pruning due to their critical impact on overall model quality. Does the author have any comments on that?*"
>
> **Response**: Thank you for the thoughtful comment and for highlighting this apparent inconsistency. Figure 1 in the motivation section reports Taylor-based importance scores on WikiText-2, and its purpose is not to claim that the first layer is universally unimportant. Rather, the figure is meant to illustrate our key observation that layer importance is highly sensitive to the specific input, and that the ranking can vary substantially across different sequences.
>
> Our decision to always retain the first and last Transformer blocks is based on two considerations: (1) Prior empirical evidence. Multiple structure pruning studies have reported that the first and last layers play disproportionately important roles in maintaining overall model quality. The first layer is crucial for early lexical transformations, while the last layer strongly influences the final logits. (2) Architectural motivation in Buddy. We reuse the first-layer KV cache to provide a stable global context representation for routing. Removing the first layer would break this mechanism. Thus, for both practical and empirical reasons, we exclude the first and last layers from pruning.
>
> To further validate these choices, we added additional analyses in Appendix A.2. Using ∆PPL-based importance on both WikiText-2 and PTB, we find that the first and last layers consistently receive high importance, aligning with prior literature. At the same time, we again observe that importance rankings vary noticeably across datasets and metrics, reinforcing the point that no single static ranking is reliable, and motivating the need for dynamic, input-adaptive pruning.

---

> ### Author Response · Authors · 2025-11-26
>
> **Questions:** "*What is the deployment cost of Buddy? For example, what is the training cost of the Decision Module? Does it only fit one model, thus individual training is required?*"
>
> **Response**: Thank you for the question. Buddy is designed to keep deployment and training cost modest. The **Decision Module is trained jointly with the base model using standard supervised fine-tuning (SFT) with LoRA**, rather than requiring a separate heavy training phase. Concretely, we fine-tune on the Alpaca instruction dataset with the usual next-token cross-entropy loss, and update both the LoRA parameters and the Decision Module parameters in a single run. In practice, this adds only a small overhead on top of a normal SFT job (the Decision Module is a lightweight MLP with negligible parameter count compared to the backbone), so the additional training cost is close to that of standard instruction tuning.
>
> Once trained for a given backbone (e.g., LLaMA-2-7B or Qwen2.5-7B), **a single Buddy model can serve all sparsity levels / budgets at inference time** via the Decision Module’s routing, and **does not need to be retrained for each target sparsity**. This is in contrast to many static pruning methods, which typically require training a separate model for every sparsity configuration. It is true that, like other adapter-style methods, Buddy needs to be trained once per underlying backbone (e.g., one run for LLaMA-2-7B, one for Qwen2.5-7B), but we view this as a one-time cost per model family; after that, the same trained Buddy model flexibly supports a wide range of user budgets without additional retraining. We report the training cost in Appendix A.4.1 of our revised version.

---

> ### Author Response · Authors · 2025-11-26
>
> **Questions:** "*Has Buddy been tested beyond the LLaMA-2/1 family to evaluate its generality? For instance, how would it perform on newer architectures like LLaMA-3 or Qwen-3, and on reasoning-intensive generation tasks such as GSM8K or MATH?*"
>
> **Response**: Furthermore, we evaluated Buddy on a more challenging, reasoning-intensive GSM8K benchmark in addition to the commonsense reasoning tasks in the main paper. Across both architectures (Llama2-7B and Qwen2.5-7B) and under multiple sparsity settings, Buddy consistently outperforms static and dynamic pruning baselines on GSM8K. These new results demonstrate that Buddy not only generalizes well to newer LLM backbones, but also maintains strong advantages on multi-step mathematical reasoning. We alseo have included the full experimental tables in the revised Appendix.
>
> | Method | pruning method | rm blocks (ratio) | Llama2-7B | Qwen2.5-7B-Instruct |
> | --- | --- | --- | --- | --- |
> | Dense w/o | - | 0(0.0%) | 14.03 | 82.87 |
> | Shortened Llama | static | 4(12.5%) | 5.16 | 27.22 |
> | ShortGPT | static | 4(12.5%) | **8.11** | 35.94 |
> | SLEB | static | 4(12.5%) | 3.64 | 37.15 |
> | PuDDing | dynamic | 4(12.5%) | 1.74 | 0.99 |
> | FiRST | dynamic | 4(12.5%) | 0.91 | 27.98 |
> | Buddy | dynamic | 4(12.5%) | 7.81 | **51.10** |
> | Shortened Llama | static | 8(25%) | 2.05 | 5.84 |
> | ShortGPT | static | 8(25%) | 2.50 | 6.60 |
> | SLEB | static | 8(25%) | 1.97 | 6.37 |
> | PuDDing | dynamic | 8(25%) | 0.99 | 0.61 |
> | FiRST | dynamic | 8(25%) | 1.21 | 2.58 |
> | Buddy | dynamic | 8(25%) | **2.65** | **14.63** |

---

> ### Author Response · Authors · 2025-11-26
>
> **Comment: “**Typos: “Softmsax” (line 244) in equation 5 should be Softmax.**”**
>
> **Response**: Thank you very much for pointing out this minor typographical error. We have corrected the misspelling in Equation (5) from *Softmsax* to *Softmax* in the revised version.

---

### Comment · Area_Chair_Kev1 · 2025-11-26
**Author-Reviewer-AC Discussion (DDL: 12/3 9PM UTC)**

Dear Reviewers,

Thank you once again for your service to ICLR 2026. Now that the authors have submitted their rebuttal, I kindly ask you to take the following steps (if you have not done so already):

- Read the authors’ response and other reviews.
- Consider whether the rebuttal and additional comments affect your assessment of the paper.
- Engage in **interactive discussion** with the authors. You may post the feedback to the authors so that they can further follow up. If you have more concerns/questions (e.g., requesting clarifications, new results), it is recommended to post your request *asap*, so that the authors have enough time to address them. **Note the Author-Reviewer-AC discussion period ends on 12/3 9PM UTC**.

The current reviews for this paper are **mixed (scores: 4/4/2/6)**. Your further contributions are essential for forming a well-informed final decision.

I am happy to join and support the discussions between you and the authors. Please feel free to share your thoughts and participate actively in the discussion. Thanks!

Best regards,

AC

---

### Meta-Review · Area_Chair_n8pm · 2026-01-05

**Summary:**

Reviewers had four main concerns: (1) novelty is limited since gating-based layer skipping has been explored before, (2) performance gains over static pruning are modest and inconsistent across sparsity levels, (3) batch processing with different routing paths per sample is impractical for real deployment, and (4) initial experiments were on outdated models.

**Reviewer Concerns:**

Addressed: Authors added LLaMA-3 and Qwen2.5 experiments showing consistent improvements. GSM8K results were included. Training procedure was clarified - Decision Module trained jointly via SFT with STE for gradient flow. Additional baselines (SkipTransformer, DiffSkip) were compared.

Outstanding: Batch processing efficiency remains unresolved - authors acknowledge this is a field-wide issue but offer no solution. Performance drop on GSM8K vs dense models is severe (51.10 vs 82.87 on Qwen). Gains over static pruning are marginal in many settings while adding significant complexity.

**Reviewer Scores:**

•	Reviewer rpEZ: Final score 2.
	•	Reviewer FF78: Final score 4.
	•	Reviewer iBu1: Final score 4.
	•	Reviewer c6ZX: Final score 4.

---

### Decision · Program_Chairs · 2026-01-26

Reject